# NAVI: Category-Agnostic Image Collections with High-Quality 3D Shape and Pose Annotations

Varun Jampani*      Kevis-Kokitsi Maninis*      Andreas Engelhardt      Arjun Karpur
Karen Truong      Kyle Sargent      Stefan Popov      André Araujo
Ricardo Martin-Brualla      Kaushal Patel      Daniel Vlasic      Vittorio Ferrari
Ameesh Makadia      Ce Liu[†]      Yuanzhen Li      Howard Zhou

Google

## Abstract

Recent advances in neural reconstruction enable high-quality 3D object reconstruction from casually captured image collections. Current techniques mostly analyze their progress on relatively simple image collections where Structure-from-Motion (SfM) techniques can provide ground-truth (GT) camera poses. We note that SfM techniques tend to fail on in-the-wild image collections such as image search results with varying backgrounds and illuminations. To enable systematic research progress on 3D reconstruction from casual image captures, we propose 'NAVI': a new dataset of category-agnostic image collections of objects with high-quality 3D scans along with per-image 2D-3D alignments providing near-perfect GT camera parameters. These 2D-3D alignments allow us to extract accurate derivative annotations such as dense pixel correspondences, depth and segmentation maps. We demonstrate the use of NAVI image collections on different problem settings and show that NAVI enables more thorough evaluations that were not possible with existing datasets. We believe NAVI is beneficial for systematic research progress on 3D reconstruction and correspondence estimation. Project page: https://navidataset.github.io

## 1   Introduction

The field of 3D object reconstruction from images or videos has been dramatically transformed in the recent years with the advent of techniques such as Neural Radiance Fields (NeRF) [43]. With recent techniques, we can reconstruct highly detailed and realistic 3D object models from multiview image captures, which can be used in several downstream applications such as gaming, AR/VR, movies, etc.

Despite such tremendous progress, current object reconstruction techniques make several inherent assumptions to obtain high-quality 3D models. A key assumption is that the near-perfect camera poses and intrinsics are given or readily available via traditional Structure-from-Motion (SfM) pipelines such as COLMAP [52]. This assumption imposes several restrictions on the input image collections. The input images have to be of sufficiently high-quality (e.g. non-blurry) and the number of input images should also be high (typically > 30-50) for SfM to estimate sufficient correspondences across images. In addition, SfM techniques typically fail on internet image sets that are captured with varying backgrounds, illuminations, and cameras. Such internet image collections do not require active capturing and are widely and readily available, such as product review photos or image search results (*e.g.*, internet images of Statue-of-Liberty, Tesla Model-3 car, etc.). It is highly beneficial to develop 3D object reconstruction techniques that can automatically produce high-quality 3D models from such image collections in the wild.

---

*Equal contribution

[†]C. Liu's current affiliation is Microsoft

37th Conference on Neural Information Processing Systems (NeurIPS 2023) Track on Datasets and Benchmarks.

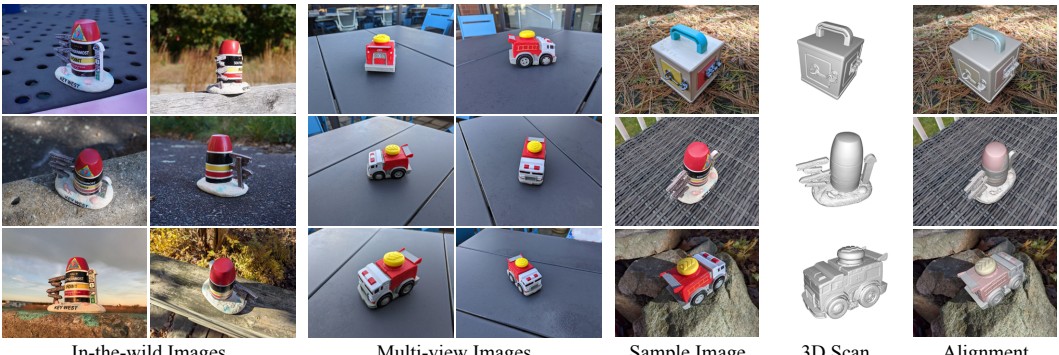

| In-the-wild Images | Multi-view Images | Sample Image | 3D Scan | Alignment |

Figure 1: **NAVI dataset overview**. NAVI dataset consists of both multiview and in-the-wild image collections, where each image is aligned with the corresponding 3D scanned model resulting in high-quality 3D shape and pose annotations.

In this work, we propose a new dataset of image collections which we refer to as 'NAVI' (Not AVerage Image dataset). Specifically, our dataset contains two types of image collections with near-perfect camera poses and 3D shapes: 1. Standard multiview object captures and 2. In-the-wild object captures with varying backgrounds, illuminations and cameras. Fig. 1 shows examples of the in-the-wild and multiview images in NAVI along with the 2D aligned 3D scans. Next, we describe the key distinguishing properties of the NAVI dataset in relation to existing datasets.

**Casual captures**. Several existing multiview datasets are either synthetic or captured in lab settings [43]. We capture NAVI images in casual real settings using hand-held cameras.

**In-the-wild image collections**. In addition to typical multiview images, NAVI also provides in-the-wild image collections where objects are captured under varying backgrounds, illuminations, and cameras. SfM techniques usually fail on such image sets and NAVI provides a unique opportunity to systematically research joint shape and camera estimation from in-the-wild image collections.

**Near-perfect 3D geometry and camera poses**. We use high-quality 3D scanners to get 3D shape ground-truth and also obtain high-quality 3D camera pose annotations with manual 2D-3D alignment along with rigorous verification. This is in contrast to several recent datasets such as [64] that rely on SfM to provide GT, thereby limiting the image capture setups.

**Accurate dense correspondences**. We provide accurate per-pixel correspondences using the 3D shape alignments. While most real-world datasets for correspondence evaluation rely on known homographies [3] or sparse keypoint annotations recovered from estimated geometry [14, 27], NAVI's precise 2D-3D alignments lead to accurate and dense object correspondences.

**Derivative annotations** such as pixel-accurate object segmentation and monocular depth can be easily derived from high-quality 2D-3D alignments in NAVI.

**Category-agnostic**. Objects in the NAVI dataset are category-agnostic with image collections that do not have any category-specific shapes, which is in contrast to widely-used 2D-3D datasets [58, 65].

To demonstrate the utility of NAVI, we benchmark and analyze some representative techniques on three different problem settings: multiview object reconstruction, 3D shape and pose estimation from in-the-wild image collections, and dense pixel correspondence estimation from image pairs. In addition to these problem settings, one could also use NAVI images for other single-image vision problems such as single image 3D reconstruction, depth estimation, object segmentation, etc.

## 2 NAVI dataset

### 2.1 Dataset construction

**Challenges**. It is worth emphasizing the challenges in our data construction by taking a look at some existing 2D-3D aligned datasets. Several works [10, 13, 29, 15, 18] propose synthetic 3D assets which are used to render 3D-aligned images. Real-world datasets such as Scan2CAD [2] and Pascal3D+ [65] use nearest intra-category CAD models for alignment w.r.t 2D images, resulting in only coarse annotations. Similarly, IKEA Objects [35] and Pix3D [58] annotate retrieved images by aligning one 3D CAD model to images using point correspondences. Even for datasets with mostly

exactly-matching products [58], slight deformations and moving parts that appear different on images with respect to their 3D scan can lead to inaccurate alignments. Different instances of the same object can also have different shapes due to other factors (e.g. shoes of different sizes are not uniformly scaled versions etc.) Fig. 2 shows the sample alignments from the existing datasets showcasing the challenges in obtaining near-perfect 3D shapes and in the 2D-3D alignment.

**Rationale**. To avoid such issues in NAVI, we selected rigid objects without moving parts, manually scanned the object shape and took image captures of *the same* objects in diverse real-world settings. We then use our interactive alignment tool to obtain near-perfect 2D-3D alignments with precise pose control during the annotation. Most datasets, including our earlier attempts, use a multi-stage alignment process that involves annotating point correspondences and then optimizing the object pose. Even though this is a more scalable approach for dataset creation, the alignments are not as accurate as we want. The NAVI dataset construction consists of 4 steps: (1) Scanning the 3D objects, (2) Capturing image collections, (3) 2D-3D alignment, and (4) Alignment verification.

**1. Scanning the 3D objects**. We collect 36 rigid objects and use two professional 3D scanners, EinScan-SP [20] and EinScan Pro HD [19], to obtain high-quality 3D object scans. We center the scans at origin, but do not normalize the shapes to preserve their metric dimensions (in mm). Fig. 3 displays some NAVI images and their aligned 3D scans. Notice the diverse and category-agnostic nature of the objects.

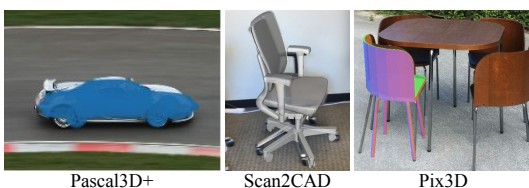

Pascal3D+  Scan2CAD  Pix3D

Figure 2: **2D-3D alignments from existing datasets** have issues as 3D models do not exactly match the corresponding 2D image due to model discrepancies.

**2. Capturing image collections**. For each object, we captured two types of image collections: in-the-wild, and multiview. In-the-wild captures contain images with different backgrounds, illumination, and cameras. Multiview captures offer the standard multiview setup: same camera, object pose, and environment, but with different camera poses. For practical utility, we captured the images in casual settings with hand-held cameras ranging from mobile phones to DSLRs and mirrorless cameras. In total, we use 12 different cameras to capture around 10.5K images with 2.3K in-the-wild images and 8.2K multiview images. More dataset details are present in the Appendix (Section A).

**3. 2D-3D alignment**. The goal is to obtain near-perfect 2D-3D alignments; *i.e.*, accurate 6DoF rigid object transformations along with accurate camera intrinsics. We developed an interactive tool on which the user can progressively align the 3D object by rotating and translating it in 3D, using the mouse. Since we know the cameras used to capture the images, we initialize the camera focal length, which can be further refined during the alignment process. Our interactive tool gives the user full control over the alignments, and we observe that this leads to higher-quality poses than alternative implicit alignment tools that optimize the pose from 3D↔2D point correspondences [58]. We trained 10 dedicated annotators for our alignment task allowing us to obtain higher quality annotations than several existing datasets that rely on generic non-expert annotators. Refer to the Appendix for more details on the alignment tool and the process (Section A.1).

**4. Alignment verification.** To ensure high-quality annotations, we further manually verify each 2D-3D alignment with 2 expert annotators. Specifically, we overlay the 3D shape onto the 2D image and ask trained annotators to label them as 'incorrect' if the alignments look even slightly wrong. For images labeled 'incorrect', we repeat the 2D-3D alignment and verification steps. After two stages of alignment and verification, we discard around 7% of the original captured images. We further annotate images with a binary occlusion label to indicate if the object is occluded by other objects. We exclude occluded object images from our validation sets for different tasks to avoid introducing artifacts in the metrics.

**Derivative annotations**. In addition to the full 3D alignments of scans to images, there are several derivative annotations that result from the accurate 2D-3D alignments: Relative camera poses, dense correspondences, metric depth maps, and binary masks. Relative camera poses are an implicit output of alignment, as all objects were posed with respect to their canonical pose. Since we have annotated multiple images of the same object, we obtain dense correspondences on the images by sampling the pixels in mutually visible parts of the 3D shape in image pairs. This enables dense correspondence evaluation both for the standard multiview setup, and for in-the-wild images captured in different environments. Fig. 4 visualizes sample GT pixel correspondences on NAVI image pairs. Furthermore,

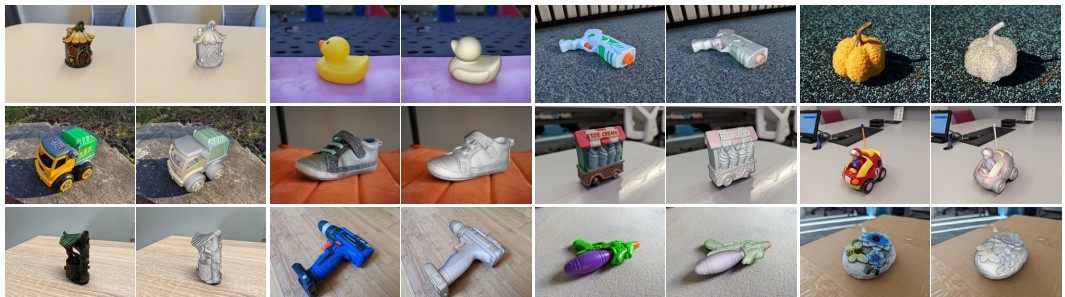

Figure 3: **NAVI samples**. Sample object images and the corresponding 2D-3D alignments. NAVI consists of casually-captured and category-agnostic image collections with precise 2D-3D alignments.

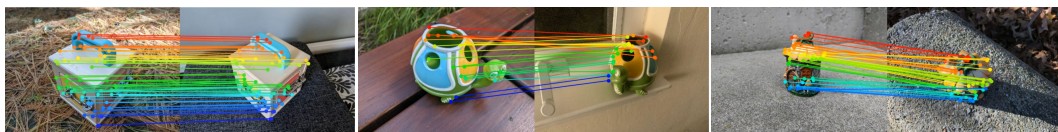

Figure 4: **Pixel correspondences**. Sample image pairs and their corresponding GT pixel correspondences. For visualization purposes, we show sparsely sampled points and color-code the correspondences based on their 2D location from top to bottom.

metric depth maps are obtained by computing the depth of the 3D alignments from the camera viewpoint. The binary object masks are trivially obtained by binarizing the depth maps. Fig. 5 shows sample object depth and mask annotations in NAVI. For simplicity, we refer to our annotations as GT.

## 2.2 Dataset analysis

**General statistics**. Table 1 presents the general statistics of the NAVI dataset. It contains 10515 alignments in total, on 36 complicated object shapes, divided into in-the-wild images (2298) and multiview images (8217). Each object is aligned on 65 in-the-wild images on average. There are 267 unique multiview scenes, some of which were also captured by different cameras (324 multiview captures in total).

**Annotation quality analysis**. To analyze the quality of our 2D-3D alignments, we annotated 30 randomly selected images with two different annotators and measured the inter-annotator agreement [30] using two metrics: 3D translation distance (in milimeters), and 3D rotation distance (in degrees) between the two alignments. The average 3D rotation distance between two verified alignments is 1.7 degrees, and the 3D translation distance is 0.97

| | |
|---|---:|
| # Alignments (total) | 10515 |
| # Alignments (wild) | 2298 |
| # Alignments (multiview) | 8217 |
| # Objects | 36 |
| # Multiview Scenes | 324 |
| # Multiview Scenes (unique) | 267 |

Table 1: **General statistics of NAVI.**

milimeteres. The very small differences in the obtained alignments from two independent raters highlight the high quality of the alignments in the NAVI dataset. Similarly, we measured the quality of the alignments that we reject as "wrong", by comparing them to alignments of the same image that were verified as "correct". In this case, average annotator disagreement is 2.3 degrees of 3D rotation, and 2.01 milimeters of 3D translation. Practically, this means that even slight mis-alignments did not pass our strict verification process. To put these numbers into perspective, the rotation error of fully automatic methods are at least one order of magnitude larger (Table 4).

**Manual vs. semi-automatic alignment process**. Other works explore the use of $3D \leftrightarrow 2D$ point correspondences for semi-automatically aligning a shape to an image [58, 39]. In this workflow the annotator needs to define points on the 3D shape and their corresponding points on the image. Alignments result from a pose optimization process. We explored such approach in the earlier stages of this work, and we argue that this process does not yield alignments as accurate as ours, mainly because the annotator does not have full control over the final alignment. For a quantitative comparison, we annotated the same 30 randomly sampled images with point correspondences, produced alignments using the software from [39], and measured inter-annotator agreement between two rounds of annotation. The average 3D rotation distance was 6.5 degrees, and the average 3D translation distance was 7.02 milimeters, significantly higher than our alignments (1.7 degrees, and

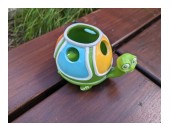 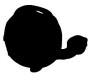 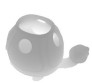 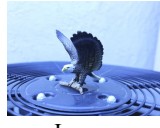 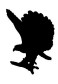 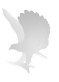

| Image | Mask | Depth | Image | Mask | Depth |

Figure 5: **Sample depths and masks**. 2D-3D alignments on NAVI images allows to readily obtain high quality object depths and mask annotations.

0.97mm, respectively). For a qualitative comparison of the outputs of the two annotation processes, please refer to Appendix (Section A.3, Figure 11).

**On using self-captured vs. Internet images**. One possible way of scaling up the annotation pipeline could be to automate the acquisition process. Instead of self-captured images that we use in this work, we experimented with Google Image Search to scrape images from the internet, and align them. We used realistic renderings of the objects in [18] as reference. We noticed that even though the poses are in general of good quality, small differences in the shape introduce significant noise to the quality of the alignments. In this work, we strive for near-perfect alignments that enable tasks very sensitive to their quality, and we thus refrain from using internet images. For an illustration of common problems faced when using internet images, please refer to Appendix (Section A.4, Figure 12).

## 3   3D from multiview image collections

**Problem setting**. Given a set of images taken from different viewpoints, the task is to reconstruct the 3D shape and appearance of an object. The 3D representation can then be used for downstream tasks like scene editing, relighting, and rendering of novel views. Traditional multiview reconstruction pipelines such as Structure-from-Motion (SfM) first reconstruct camera poses together with a sparse object representation followed by a dense reconstruction and potential mesh generation step. After adding materials and textures, the resulting 3D asset can then be used to render new views. More recent techniques such as NeRF [43] optimize neural representations of objects directly on the RGB images with the camera poses obtained from an SfM reconstruction as a pre-processing.

**Related datasets**. Synthetic multiview scenes [56, 43, 25] are widely adopted for evaluations. In contrast to synthetic scenes that come with precise 3D scene and camera poses but only translate to real-world photography to a limited degree, real scenes usually require off-the-shelf SfM methods [52] for pose estimation. BlendedMVS [70], one of the first multi purpose datasets for stereo reconstruction comes with re-rendered images based on geometry and poses reconstructed via a SfM pipeline. CO3D [48] and Objectron [1] are large-scale datasets with object-centric videos, and provide either a rough point cloud reconstruction of the object [48] or a 3D bounding box [1]. The dataset of [28] offers a handful of 3D laser scans along with the corresponding real-world image collections. Recently, works of [55] and OmniObject3D [64] provide 3D object scans along with multiview image captures in constrained lab settings. These works rely on SfM for semi-automatic 2D-3D alignment. In summary, existing multiview datasets are synthetic [56, 43, 25] or based on reconstructed 3D models [70], with rough 3D shapes [48], provide only a limited number of scenes [28] or they consist of image captures in constrained settings [55, 64].

**The distinctiveness of NAVI**. In contrast, NAVI satisfies multiple requirements by offering highly-accurate 3D shapes and alignments for multiple objects from different categories in different real-world environments and illumination. This allows for more precise evaluation of 3D reconstruction techniques on real-world object image collections.

**NAVI dataset and metrics**. We split each of the multiview image sets into 80%/20% train/validation sets. The multiview sets are object-centric with an average of 25 images per set (minimum 3 to maximum 180). For even evaluation across the objects, we randomly sample 5 multiview scenes for each object from the subsets that include more than 6 images, resulting in 180 multiview sets for our experiments. We use the standard novel view synthesis metrics, PSNR, SSIM and, LPIPS [74], on validation images and report average metrics across all sets.

**Experiment**. A key assumption in most existing works is that SfM provided camera poses are good enough for 3D reconstruction. We want to test this hypothesis by evaluating how our annotated camera poses compare against COLMAP [52] poses for off-the-shelf 3D reconstruction techniques. For this, we use the widely-used InstantNGP [45] to reconstruct Radiance Fields from multiview images. For optimization we use the GT masks to limit the reconstruction to the object area.

| Dataset | # Objects | # Scenes | # Images | Camera Poses | 3D Annotations | 3D↔2D Alignment |
|---|---|---|---|---|---|---|
| LASSIE [68] | 6 | 6 | 180 | - | Keypoints | ✗ |
| E-LASSIE [69] | 6 | 6 | 270 | - | Keypoints | ✗ |
| NeRD [6] | 8 | 8 | 396 | Synthetic-GT | - | ✗ |
| NeRF-W [40] | - | 6 | 7658 | - | - | ✗ |
| SAMURAI [8] | 8 | 8 | 560 | - | - | ✗ |
| NeROIC [31] | 3 | 3 | 132 | COLMAP | - | ✗ |
| NAVI (Ours) | 36 | 36 | 2298 | Near-GT | Scanned mesh | ✓ |

Table 3: **Comparing NAVI with existing in-the-wild image collections,** where the task is to 3D reconstruct an object given its images captured in different environments and lighting settings.

**Results: COLMAP vs. GT poses**. Table 2 shows the novel view synthesis metrics on validation images. Results on all the metrics demonstrate considerably better reconstruction with our GT poses compared to using COLMAP poses. COLMAP only registers partial set of views for several cases. This shows that the our GT poses are accurate and are still valuable in the multiview reconstruction setting to analyze reconstruction techniques independent of inaccuracies from the camera registration. While COLMAP poses are arbitrarily rotated and scaled, all NAVI scenes are centered at the origin and in a common coordinate frame. This facilitates evaluation across different objects, especially in the context of grid-based methods like InstantNGP where the scene bounds have some impact on performance.

## 4   3D from in-the-wild image collections

**Problem setting**. The aim is to estimate 3D shape and appearance of an object given an *unconstrained* image collection; where the object is captured with different backgrounds, cameras and illuminations. Such image collections are readily available on the internet; *e.g.*, image search results, product review photos, etc. The high variability in the appearance across images makes pose estimation and reconstruction highly challenging compared to the more controlled multiview captures. Techniques

| Camera Poses | PSNR↑ | SSIM↑ | LPIPS↓ |
|---|---|---|---|
| COLMAP | 24.04 | 0.93 | 0.079 |
| NAVI Poses (GT) | 27.54 | 0.94 | 0.045 |

Table 2: **View synthesis metrics using COLMAP and GT poses.** Our GT poses lead to significantly better performance, because COLMAP [52] does not always accurately pose our multiview scenes.

need to jointly reason camera poses and illuminations in addition to 3D geometry and appearance. Standard SfM techniques [53, 52] fail to recover camera poses on such in-the-wild image sets.

**Existing datasets**. Curated object centric image collections from in-the-wild data are scarce. While one could search online image databases for multiple occurrences of the same object or class [68], additional data like camera parameters or object shape as well as the certainty that all images actually depict the same object instance is critical for faithful evaluation. Table 3 compares published image collections used for 3D reconstruction from in-the-wild data to NAVI. Most existing collections include only a few number of scenes and objects and, if at all, sparse 3D annotations. NAVI is the only dataset of that kind with real-world image collections of objects in the wild (with varying environments and cameras), and with near-perfect 2D-3D alignments of 3D meshes. Other datasets such as Pix3D [58] contain many images (10k) with aligned 3D objects, which are however too inaccurate to be used for the task. DTU MVS dataset [26] is widely used as a proxy for in-the-wild data [60, 41, 71] as it comes with different lighting conditions for each of the 124 scenes. However, the controlled acquisition environment does not fully reflect in-the-wild conditions. Additionally, 3D scans and depths are of limited quality and coverage since the structured-light scan is only acquired at the given view positions. NeROIC [31] and NeRD [6] provide small collections of scenes for 360° object reconstruction featuring lighting changes and poses reconstructed via SfM. However, no GT object shapes are included. SAMURAI [8] adds eight image sets to the NeRD dataset with different cameras, backgrounds and illuminations; but it only provides RGB images without any GT camera poses or shapes. NAVI dataset subsumes these 8 SAMURAI in-the-wild image collections where we provide near-GT poses and 3D shapes.

**The distinctiveness of NAVI**. NAVI provides the first real-world in-the-wild image collections with GT 3D shapes and camera poses. For evaluation, existing techniques such as [8] rely on novel view synthesis metrics on held-out images which entangle the role of estimated camera poses and shapes. It is not possible to assess whether the view synthesis is poor due to a wrongly estimated camera pose or a wrongly estimated 3D object. GT poses and shapes in NAVI wild-sets provide a unique

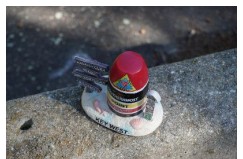 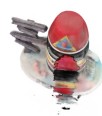 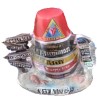 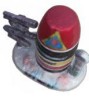

(a) GT Novel View Image     (b) NeROIC     (c) NeRS     (d) SAMURAI

Figure 6: **Novel view synthesis with in-the-wild 3D reconstruction**. Sample novel view synthesis results of different techniques for 3D reconstruction from in-the-wild image collections.

opportunity to systematically analyze different techniques using pose metrics. In addition, NAVI also enables thorough analysis of techniques with controlled noise levels in the camera parameters.

**NAVI dataset and metrics**. We divide each of the in-the-wild image sets of NAVI into $80\%$ / $20\%$ splits for training and validation respectively, where the techniques optimize a 3D asset using the train images and are evaluated on validation sets. On average there are 65 images in each in-the-wild set with minimum of 46 and maximum of 93 images, respectively. We use 2 different setups for evaluation. First is the standard novel view synthesis metrics that measure PSNR, SSIM and LPIPS [74] scores on held-out validation images. Second is camera pose evaluation where we use Procrustes analysis [24] to compute the mean absolute rotation, translation and scale difference in camera pose estimations for all the images. The camera metrics are a unique feature of NAVI enabled by our near-GT poses, compared to existing real-world datasets with in-the-wild image collections.

**Techniques**. We analyze four recent reconstruction techniques that can jointly optimize camera poses and can also deal with varying illuminations to some extent: NeRS [73], SAMURAI [8], NeROIC [31], and GNeRF [41]. Different works use different camera initialization and also model the object appearance differently. NeROIC assumes roughly correct COLMAP poses. NeRS and SAMURAI assume rough quadrant pose initialization and GNeRF takes randomly initialized poses. See Appendix (Section C.1) for a brief introduction of these techniques and refer to their respective papers for more details. While these techniques use either pre-computed or GT objects masks, we use GT object masks in our experiments to ensure fair comparison.

## 4.1 Analysis

**COLMAP vs. GT poses**. Table 4 shows the view synthesis performance and camera pose errors for different techniques and camera initializations. We observe that COLMAP reconstruction only works for a subset of scenes $S_C$ (19 out of 36 scenes) for which the camera pose estimation using COLMAP yields more than 10 cameras. For comparisons with NeROIC that rely on COLMAP initialization, we separately report the metrics on scenes $S_C$ where COLMAP works and those where COLMAP fails ($\sim S_C$). We omit one scene (vitamins bottle) that shows some inconsistencies between views because of a moving cap. Compared to the results from Section 3, the increased complexity of the task is reflected in lower performance. Comparing the performance of NeROIC with COLMAP to the initialization with NAVI GT poses on the $S_C$ subset, it is clear that the NAVI GT poses are also superior in this setting. In addition to any COLMAP inaccuracies, the 3D reconstruction task becomes harder as the number of images shrinks due to incomplete COLMAP pose recovery that recovers only a subset of views.

| Method | Pose Init | PSNR↑ | | SSIM↑ | | LPIPS↓ | | Translation↓ | | Rotation °↓ | |
|---|---|---|---|---|---|---|---|---|---|---|---|
| | | $S_C$ | $\sim S_C$ | $S_C$ | $\sim S_C$ | $S_C$ | $\sim S_C$ | $S_C$ | $\sim S_C$ | $S_C$ | $\sim S_C$ |
| NeROIC [31] | COLMAP | 19.77 | - | 0.88 | - | 0.1498 | - | $0.09 \pm 0.12$ | - | $42.11 \pm 17.19$ | - |
| NeRS [73] | Directions | 18.67 | 18.66 | 0.92 | 0.93 | 0.1078 | 0.1067 | $0.49 \pm 0.21$ | $0.52 \pm 0.19$ | $122.41 \pm 10.61$ | $123.63 \pm 8.80$ |
| SAMURAI [8] | Directions | 25.34 | 24.61 | 0.92 | 0.91 | 0.0958 | 0.1054 | $0.24 \pm 0.17$ | $0.35 \pm 0.24$ | $26.16 \pm 22.72$ | $36.59 \pm 29.98$ |
| GNeRF [41] | Random | 8.30 | 6.25 | 0.64 | 0.63 | 0.52 | 0.57 | $1.02 \pm 0.16$ | $1.04 \pm 0.09$ | $93.15 \pm 26.54$ | $80.22 \pm 27.64$ |
| NeROIC [31] | GT | 22.75 | 21.31 | 0.91 | 0.90 | 0.0984 | 0.0845 | $0.07 \pm 0.24$ | $0.01 \pm 0.01$ | $33.17 \pm 19.63$ | $31.90 \pm 11.11$ |
| NeRS [73] | GT | 17.92 | 18.02 | 0.92 | 0.93 | 0.114 | 0.1098 | $0.62 \pm 0.19$ | $0.65 \pm 0.20$ | $86.96 \pm 27.63$ | $89.43 \pm 22.60$ |
| SAMURAI [8] | GT | 25.65 | 25.59 | 0.92 | 0.92 | 0.0949 | 0.0881 | $0.16 \pm 0.14$ | $0.25 \pm 0.26$ | $21.55 \pm 21.72$ | $28.25 \pm 26.71$ |

Table 4: **Metrics for 3D shape and pose from image collections in the wild.** View synthesis and pose metrics over two subsets from all wild-sets depending on the success of COLMAP ($S_C$ / $\sim S_C$). Rendering quality is evaluated on a holdout set of test views that are aligned as part of the optimization without contributing to the shape recovery. We include GNeRF as a separate baseline although this method is not designed for multi-illumination data. We report metrics with the methods' default camera initialization as well as initializing with the GT poses that come with NAVI.

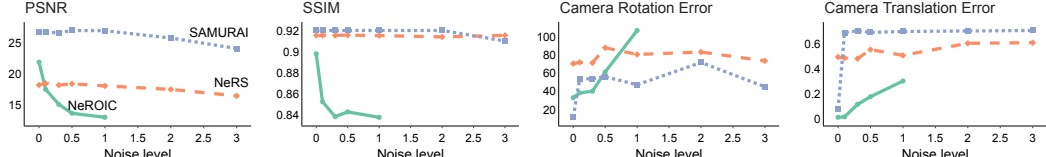

Figure 7: **Analysis with varying camera noise**. For different techniques, we initialize cameras with different levels of noise added to the GT poses for in-the-wild sets. To limit the computation, we report the mean over a subset of four objects of medium complexity.

Optimizing with GT poses can give insights into the additional challenges of the in-the-wild task independent of any dependency like COLMAP. This enables us to observe the other limitations that have an impact on in-the-wild reconstruction quality like the illumination model in SAMURAI or material model in NeRS.

**Comparing different methods**. Table 4 shows that SAMURAI performs best although the camera reconstruction quality varies drastically from scene to scene as can be seen in the large uncertainty. This is partly by design as views with large reconstruction errors are discarded over the course of optimization in this approach. It should be noted that data similar to NAVI guided SAMURAI's design. The results indicate that there are aspects covered by this data not available in other datasets (predominantely synthetic) used for evaluations so far. Fig. 6 shows sample novel view synthesis results of different techniques on an example from the "Keywest showpiece" validation set. This is a challenging object with high frequency details (e.g. text), some symmetry, and glossy surface areas. We can observe different artifacts characteristic for the evaluated methods like the rotated view and the high specularity in NeRS, texture smoothness in SAMURAI, and floater artifacts in NeROIC. NAVI includes several challenging objects that are well suited to evaluate the methods' limits. Section C.2 (Appendix) provides further results and analysis of the distribution of the reconstruction errors over different objects in the dataset.

**Camera metrics**. Thanks to the GT camera pose annotations, both the novel view synthesis and camera evaluations can be done on the same data where multiple datasets, often including synthetic data had to be used in the past. Together with the GT masks from NAVI all the confounding varying assumptions on the input data across different techniques can be made uniform here. For all the techniques, camera errors are relatively high overall, still there is a correlation between pose error and view synthesis quality. NeRS shows a surprisingly large camera pose error. It can be visually confirmed that test views are not that well aligned, still 3D mesh generation based on the training views works relatively well. Camera pose not being a focus in the original work, techniques like NeRS can benefit from explicit pose evaluations for technical improvements.

**Analysis with varying camera noise**. Annotated camera parameters in NAVI allow for a controlled study of how different techniques work with increasing amount of camera noise in their camera initialization. Specifically, we add normal distributed noise with zero mean and varying standard deviation to the annotated poses before feeding it as input to different techniques. The rotational change is limited to +/- 90° and the translation noise scales with the mean distance of the cameras to the object. A noise level of 1.0 translates to a standard deviation of 10% of the mean distance for translation and 18° standard deviation for the rotation noise on a linear scale. Fig. 7 shows the plots with novel view synthesis and camera metrics for SAMURAI, NeRS and NeROIC. While the pose error generally increases as the noise level increases, the camera rotation error is not strictly monotonically increasing, for example. This points to the shape of the loss landscape with local minima. Both SAMURAI and NeRS seems relatively robust with varying camera noises, while NeROIC performance degrades with increasing camera noise. SAMURAI seems to be robust to large noise levels but, except for GT poses, yields a high translation error. This might stem from the camera multiplex initialization and view weighting scheme. Translation can also be approximated by a focal length change to some extend which could also happen in SAMURAI where the global scene bound is part of a regularization that prefers cameras around the mean radius. NeROIC performs very well under small noise levels but cameras rotate too far away from the object bounding box for higher camera noise levels. It seems like small rotation errors can be compensated by the neural network (if conditioned on direction) to some extent here. In summary, different methodologies seem to be needed for different strengths of camera noise. NAVI can help systematically investigate how the camera optimization performs in a technique thereby informing on several useful design choices for technical improvements (e.g. larger vs. smaller pose updates, regularization weights, initialization and fine-tuning). In addition, investigations around the breaking point of a method can lead to valuable insights into the task of joint shape and camera optimization.

| Method | Correspondence Metrics | | Relative Pose Metrics | | |
| --- | --- | --- | --- | --- | --- |
| | Precision@0.2↑ | Dense-Recall@15px↑ | AUC@5°↑ | AUC@10°↑ | AUC@20°↑ |
| SIFT + MNN | 2.8 / 1.2 | 6.3 / 2.2 | 7.3 / 5.9 | 13.6 / 11.9 | 23.5 / 23.0 |
| SIFT + NN-Ratio | 10.2 / 4.8 | 6.4 / 2.2 | 6.2 / 4.2 | 11.9 / 8.1 | 22.7 / 23.1 |
| SuperPoint + MNN | 6.1 / 3.3 | 9.7 / 5.6 | 10.0 / 8.2 | 19.2 / 16.0 | 31.6 / 28.0 |
| SuperPoint + NN-Ratio | 24.7 / 18.7 | 11.4 / 6.8 | 9.0 / 7.6 | 17.5 / 15.0 | 29.0 / 26.5 |
| SuperPoint + SuperGlue | 26.8 / 23.8 | 12.6 / 9.1 | 12.1 / 10.8 | 22.2 / 20.1 | 34.6 / 32.3 |
| LoFTR | 19.2 / 13.4 | 16.3 / 8.5 | 12.2 / 9.8 | 22.5 / 18.4 | 34.2 / 30.0 |

Table 5: **Correspondence and relative pose estimation.** For each metric, we present multiview (left) / in-the-wild (right) results. To calculate precision, we filter the matching confidence of correspondences at 0.2 [57]

and use a reprojection error threshold of 3 pixels. For all other metrics, we consider the entire prediction set.

## 5    Correspondence estimation

**Problem setting**. Given a pair of images of the same object, the goal of correspondence estimation is to match a set of object pixels from one image to the corresponding pixels in the second image. By definition, an image point can have at most one correspondence in the other image as some points may be unmatched due to occlusion. Image pair correspondences are fundamental for the downstream tasks of 3D reconstruction and pose estimation, where a robust estimator is often used to recover the underlying relative camera rotation and translation.

**Existing datasets**. Finding a suitable dataset for training and evaluating correspondence estimation methods can be a challenge. SPair-71k [44] and CUB [61] provide in-the-wild semantic correspondences, but these correspondences associate parts of different objects and have limited use in instance-level tasks. Manually labeling fine-grained, instance-level correspondences is a time-consuming and error-prone task, so datasets must rely on either known real [3] or synthetic [16] homographies, or complete scene information [14, 27, 34, 54]. However, synthetic homography pairs suffer from unrealistic image distortion, and many of the latter datasets focus only on indoor/outdoor scenes and not object-centric imagery. Alternatively, high-quality 3D models [18] can be used to render object-focused image pairs with known correspondences, but methods may suffer from a wide domain gap when transferring knowledge from synthetic renderings to real world scenes.

**The distinctiveness of NAVI**. In contrast, the NAVI dataset annotations allow us to generate real-world image pairs with dense per-pixel correspondences, due to the precise 2D-3D alignments. This provides a unique opportunity to have novel *dense* evaluation metrics for correspondence estimation techniques. Additionally, the NAVI *in-the-wild* collections allow correspondences to be annotated across images with different backgrounds, lighting conditions, and camera models. For example, Fig. 4 shows sample pixel correspondences on NAVI *in-the-wild* image pairs.

**NAVI dataset and metrics**. We sample two types of correspondence datasets in NAVI. The first dataset contains randomly sampled image pairs *within the same multiview set* to represent the scenario of a fixed scene and camera model. The second dataset contains randomly sampled pairs *from the in-the-wild set* to emulate the variety of backgrounds, illuminations, and cameras. For each image pair, we can use the complete camera-object knowledge to label ground truth correspondences between the two images while respecting self-occlusions. We sample up to 707 multiview pairs and 1035 in-the-wild pairs per object resulting in the validation sets with 24745 and 35931 pairs, respectively. We limit GT correspondence labels to object pixels, since the data annotation process limits the available depth information to object points only. Additionally, we resize each image before evaluation such that their largest dimension is 1200 pixels.

For benchmarking, we evaluate both correspondence and pose estimation metrics. We use precision (reprojection error less than 3 pixels) and recall to directly evaluate correspondences, but we define a recall metric that leverages the dense ground truth correspondences made available by the NAVI 2D-3D alignment. For each object pixel visible in the first image, we find the corresponding location in the second image, after filtering out instances of self-occlusion. Given a correspondence prediction set, we calculate the percentage of ground truth matches which have a corresponding prediction whose keypoints are within N pixels of error. We denote this metric *dense recall*, and it provides an understanding of how well-distributed the predicted correspondences are across the co-visible regions. In addition, we estimate relative camera poses from the predicted correspondences. We compute the essential matrix that relates a pair of cameras using their intrinsic parameters, and the correspondences (standard 5-points algorithm). Finally, we calculate the rotation error between the

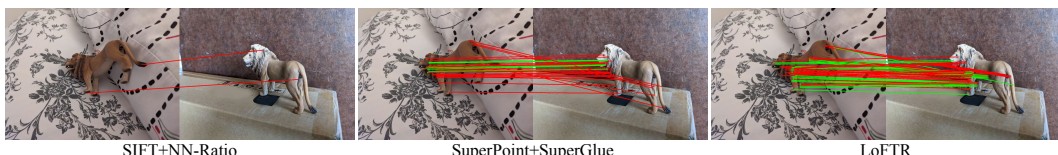

| SIFT+NN-Ratio | SuperPoint+SuperGlue | LoFTR |

Figure 8: **Sample correspondence results** of different techniques where the correct (within 3 pixels) and incorrect matches are shown in green and red respectively.

predicted and ground truth rotation matrices using Rodrigues' formula, and report accuracy within $5°$, $10°$, and $20°$ of error following [51].

**Techniques**. We evaluate the following 4 types of correspondence estimation methods: SIFT + MNN/NN-Ratio [37] that use traditional keypoint detection with heuristic traditional matching; SuperPoint + MNN/NN-Ratio [17] that use learned keypoint detection with traditional matching; SuperPoint + SuperGlue [51] that use both learned keypoint detection and learned matching and; LoFTR [57] that proposes dense learnable matching. We directly evaluate these off-the-shelf models trained on their respective datasets. See Section D.1 (Appendix) for a brief summary of these techniques and refer to the original papers for more details.

### 5.1 Analysis

**Multiview vs. In-the-wild pairs**. Table 5 presents the evaluation metrics on the *multiview/in-the-wild* image pair datasets in NAVI. Across all metrics, we observe a significant decrease in performance from *multiview* to *in-the-wild* pairs. Traditional methods (i.e. SIFT+MNN/Ratio) are insufficient to handle major changes in lighting conditions, such as ambient lighting and shadows produced by the environment. Learned methods (SuperPoint and SuperGlue) are more robust to changes across *in-the-wild* images with different backgrounds, lighting and cameras. We note that SuperGlue experiences a $3\%$ decrease from *multiview* to *in-the-wild* in Prec@0.2 and a $3.5\%$ decrease in Dense-Recall@15px, compared to $6\%$ and $4.6\%$ for the traditional matcher (SuperPoint + NN-Ratio). We also note that LoFTR proves to be less robust to changes in lighting conditions than the sparse feature-based SuperPoint+SuperGlue method. These results emphasize the importance of exposing learnable features and matchers to sufficient *in-the-wild* image pairs during training.

**Dense coverage**. Table 5 also shows *dense recall* metric enabled by dense GT correspondences in NAVI. This measures the coverage of pixel correspondences given a wide error tolerance (15 pixels). Local feature techniques are highly dependent on texture-rich regions and suffer from low coverage over smooth/textureless overlapping regions. LoFTR, a dense learnable matcher, performs well on the *multiview* split but is outperformed by SuperPoint+SuperGlue on the *in-the-wild* split. This *dense recall* metric highlights that existing matching techniques recover correspondence sets with low coverage of overlapping object regions, and that the NAVI dataset may serve as a benchmark for this important evaluation metric. Finetuning these methods on object-centric data is likely to yield better performance. Figure 8 shows some sample visual results of correspondences with different techniques.

## 6 Conclusion and discussion

**Use of NAVI in other tasks**. In addition to 3D from image collections and correspondence tasks, NAVI can be useful for single-image tasks such as single image 3D reconstruction, monocular depth or normal estimation and object segmentation. There exist several large-scale datasets for these tasks and NAVI can be used as an additional fine-tuning or evaluation dataset. We present some single image 3D reconstruction experiments in Section E.

**Limitations**. Scale is the main limitation of the NAVI dataset which consists of only 36 objects and $\approx$10K images. We prioritize annotation quality over quantity; and our current rigorous data capture and annotation pipeline is not easily scalable to collect large datasets. Since the techniques for 3D from image collections usually optimize the 3D models within an image collection, we do not find the small scale of NAVI to be a limiting factor. In the future, we also plan to extend the dataset to videos.

**Concluding remarks**. In summary, we propose NAVI dataset with multiview and in-the-wild image collections annotated with near-perfect 3D shapes and camera poses. We demonstrated the use of NAVI for better analysis on 3D from multiview image collections, 3D from in-the-wild image collections and pixel correspondence estimation problems. We believe NAVI is beneficial for a multitude of 3D reconstruction and correspondence tasks.

**Acknowledgements**. We thank Prabhanshu Tiwari, Gourav Jha, Ratandeep Singh, and Mohd Adil for coordinating the annotation process, and all annotators who contributed to NAVI. We also thank Mohamed El Banani and Amit Raj for their valuable feedback on the manuscript.

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

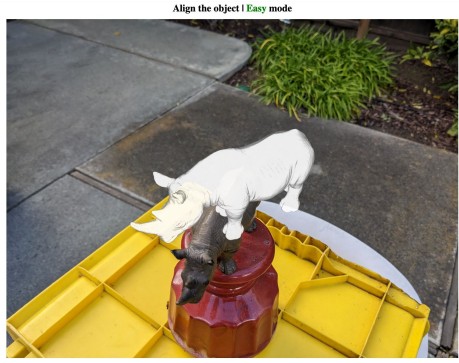
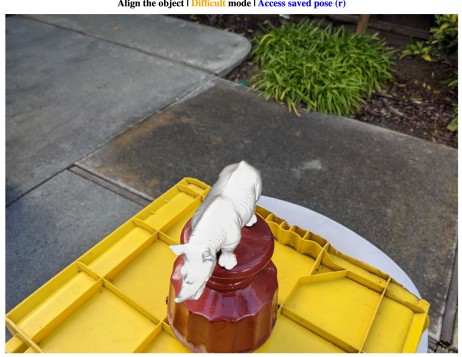

| Alignment initialization | Alignment result |

Figure 9: **3D-2D alignment tool**. The user is able to rotate and translate the 3D object directly on the screen. We provide ways to improve the annotation experience such as a restricted 'easy' mode when the object appears upright, saving a backup pose that can be recovered, enabling/disabling texture, and others.

# A    Additional dataset details

## A.1    3D↔2D alignment tool

Our interactive alignment tool was developed using the three.js library [62] that allows the user to interact with a 3D object directly on the browser. The objective is to directly produce alignments on images by rotating and translating the object.

For rotating the object, we used the intuitive mouse movements of Orbit Controls [36] that allow the user to rotate an object in 3D using 2D drag-and-drop movements. Orbit Controls constrains the 'up'-axis of the 3D shape ('easy' constrained mode), which makes the task much easier when the objects are in 'upright' position on the images. For adjustments, and for objects that do not appear in 'upright' position, the user has the option to remove the 'up-axis' constraint, and allow for all possible 3D poses ('difficult' unconstrained mode). Our tool has the option to switch between the two modes, which enables the user to first bring the object to a close-enough pose using 'easy' mode, and then switch to 'difficult' mode for the final adjustments. For translating the object, we used the panning functionality of [36].

Taking into account feedback from the annotators, we developed the option to save a backup pose, and revert back to it in case they need to restart the process (eg. when the backup pose is better than the current pose). We further developed simple keyboard shortcuts that improve the annotation experience, such as disabling/enabling texture on the 3D shape, and changing its opacity. Figure 9 illustrates our annotation interface. The camera parameters (focal length) are initialized from the ExIF metadata of the images, and can also be adjusted from within the tool.

We observed that our interactive tool gives the user full control over the alignments, and lead to higher-quality poses than alternative implicit alignment tools that optimize the pose from 3D↔2D point correspondences [58]. This allowed us to obtain annotations of higher quality than existing datasets that use general crowd-sourcing.

## A.2    Quality of 3D scans

The 3D scans were obtained by using [20] and [19]. We used their fixed scan mode (accuracy of 0.05mm and 0.04mm, respectively). Additionally, we manually curated the mesh, removed noisy vertices, and posed them in a consistent pose (Y+ axis is up, Z- axis is front, like in [10]). Figure 10 illustrates the quality of our scans. We provide accurate shapes with very fine-grained details.

## A.3    Visualization: Manual vs. semi-automatic alignment process

We provide qualitative evidence of the analysis in Section 2.2 of the main paper (paragraph 'Manual vs. semi-automatic alignment process'). Figure 11 compares our pipeline to the output of [39],

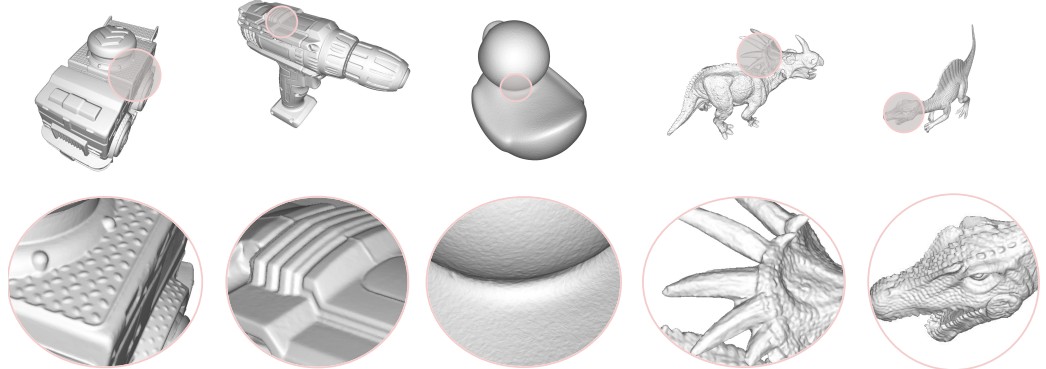

Figure 10: **Quality of 3D scans.** We provide detailed 3D meshes as a result of the scanning process.

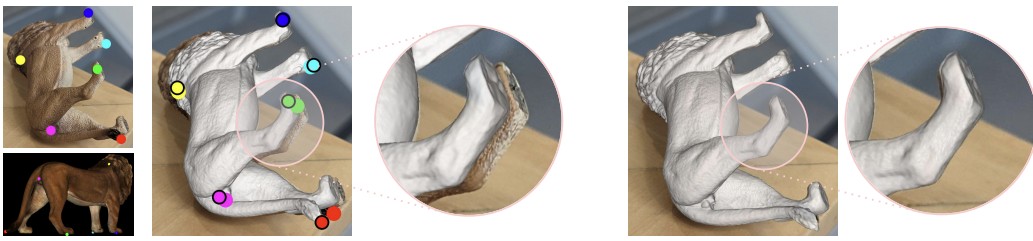

3D↔2D correspondences     Alignment from 3D↔2D correspondences     Alignment from the NAVI interactive tool

Figure 11: **Comparison to aligning from 3D↔2D correspondences**. The semi-automatic alignment process of [39] does not lead to the accuracy that we achieve in NAVI. See the mis-aligned paw and head of the lion from point correspondences (left). In our tool the annotators have full control of the final alignment and produce more accurate results (right).

a common semi-automatic alignment pipeline from 3D↔2D point correspondences. In general, we observed that [39] can not produce the near-perfect alignments that our dataset provides (eg. mis-aligned paw and head of lion from point correspondences as a result of the optimization process).

## A.4 Visualization: On using self-captured vs. Internet images

Figure 12 illustrates common issues when aligning on internet images instead of our self-captured images. While our tool can yield accurate poses on internet images, discrepancies in the shapes introduce significant noise in the quality of the resulting alignments. Discrepancies include additional/missing parts on the shapes (first, and last alignment), moving parts (second alignment), and differences in the manufacturing process (different colors in the first, and different thickness on the wheels of the third alignment). Internet images would make sense for creating a lower-quality, noisier dataset than NAVI. Nearly none of the produced alignments would pass NAVI's strict quality threshold.

| Camera type | # Images |
|---|---|
| pixel_5 | 2040 |
| pixel_6pro | 1688 |
| pixel_4a | 1674 |
| canon_t4i | 1237 |
| ipad_5 | 1140 |
| pixel_4xl | 1024 |
| pixel_7 | 869 |
| sony_a7iv | 228 |
| iphone_7plus | 224 |
| sony_a6000 | 216 |
| ipad_4 | 175 |

Table 6: **Camera types used for the images of NAVI.**

## A.5 Data license, access details, and intended usage

The dataset is released under the CC-BY license. The accompanying code that shows the dataset use is released under the Apache License 2.0. The authors of the dataset bear all responsibility in case of violation of rights.

All contents of this submission (code, paper, data) can be accessed from our project page: https://navidataset.github.io. The data is hosted on Google Cloud by Google Research. The authors will maintain and update the dataset.

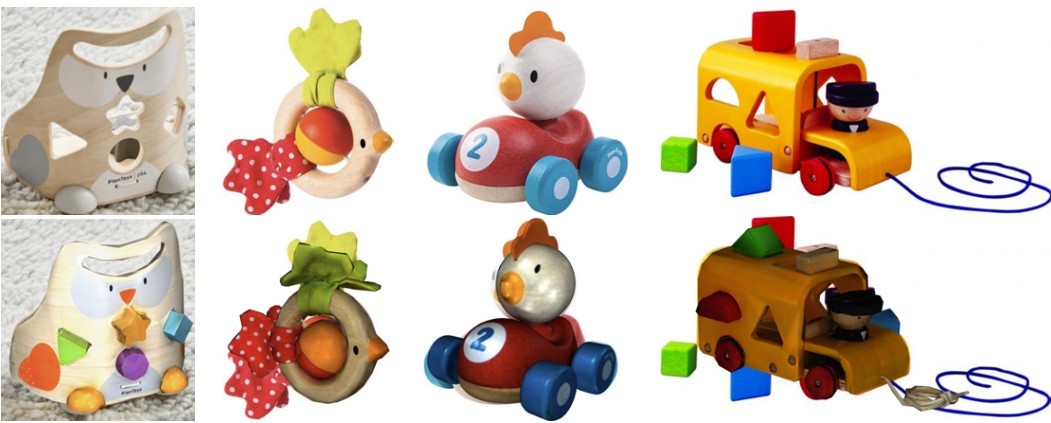

Figure 12: **Common problems when aligning on Internet images**. Small differences on the shapes introduce significant noise to the alignments. Additional parts (first, second, and last pairs), and slight differences in the manufacturing process (different colors on the first, and thicker wheels, texture mis-alignment on the third pair) are very common issues when aligning from internet images.

Extensive documentation of the dataset and how to use it for the various tasks can be found in the accompanying Github repo: `https://github.com/google/navi`. Users are invited to use the included Jupyter notebook `NAVI Dataset Tutorial.ipynb` for a quick start.

The released dataset consists of multiple folders with images (`jpg`), scans (`obj`, `mtl`, `glb`) and annotations (`json`) that connect them. Users can download the dataset at `https://storage.googleapis.com/gresearch/navi-dataset/navi_v1.tar.gz` (29GB).

The intended usage of the dataset is to enable benchmarking and systematic development of the 3D vision tasks presented in the main paper as well as this supplemental: 3D from multiview image collections, 3D from in-the-wild image collections, correspondence estimation, and 3D from a single image.

## B    3D from multiview image collections

### B.1    Comparison to existing multiview (MV) datasets

Table 7 compares NAVI (multiview part) to existing datasets of real images. While there exist datasets with much larger number of images, NAVI uniquely combines three different aspects: Very detailed 3D annotations, very high quality camera poses, and diversity of environments for a single object.

Regarding the type of 3D annotations, Objectron [1] is a large dataset that is annotated with 3D boxes. Some datasets [48, 72, 26] provide point clouds that cover parts of the objects, meshes that result from Multi-View Stereo (MVS) [70], or depth maps [9]. Closer to our work, [18] and [28] contain high quality scans resulting from a structured light scanner, and a laser scan, respectively. However, these works do not provide 2D-3D alignments of objects in real images. Regarding the quality of camera poses, most works use Structure-from-Motion (SfM) techniques like [52] for their camera poses [43, 1, 48, 26, 72, 70]. YCB [9] provides ground-truth quality poses from an RGB-D camera but was captured solely in a lab environment. Regarding diversity of environments, NAVI is the only dataset that includes scenes shot with different camera models (smartphones, consumer grade point-and-shoot, professional grade DSLR cameras, See Table 6) where the exact same object instance is shown in different illumination settings.

NAVI combines all three characteristics, and offers full 360° coverage of all objects in real environments together with high-quality 2D-3D alignments to detailed object scans. This makes it an ideal dataset for evaluating models trained on large datasets, and for training/evaluating scene-based optimization techniques such as methods of the NeRF [43] family.

Table 7: **Comparison with real multiview (MV) datasets.** We compare statistics for existing real multiview datasets. NAVI (multiview part) provides high quality 3D scans and alignment for objects captured in diverse environments using different camera models.

| Dataset | # Images | Type of Data | Camera Poses | Multi Camera | Multi Environment | 3D Annotations | 3D↔2D Alignment |
|---|---|---|---|---|---|---|---|
| NeRF LLFF [43] | 288 | MV images | SfM [52] | ✗ | ✗ | - | ✗ |
| Objectron [1] | 4M | videos | AR session | ✓ | ✗ | 3D box | (✓) |
| CO3D [48] | 1.5M | 360° videos | SfM [52] | ✗ | ✗ | point cloud | - |
| DTU MVS [26] | 40.1k | MV images | SfM | ✗ | ✓ | point cloud | ✓ |
| MVImgNet [72] | 6.5M | 180° videos | SfM [52] | (✓) | ✗ | point cloud | ✓ |
| Blended MVS [70] | 17.8k | MV images | SfM | ✗ | ✓ | MVS mesh | ✓ |
| YCB Object Set [9] | 46,2k | 360° RGB-D | GT | ✗ | ✗ | depth | ✓ |
| Tanks&Temples [28] | 153,4k | 360° videos | - | ✗ | | laser scan | ✗ |
| GoogleScan [18] | 100k | - | - | - | - | scanned mesh | - |
| NAVI (Ours) | 8217 | 360° images | Near-GT | ✓ | ✓ | scanned mesh | ✓ |

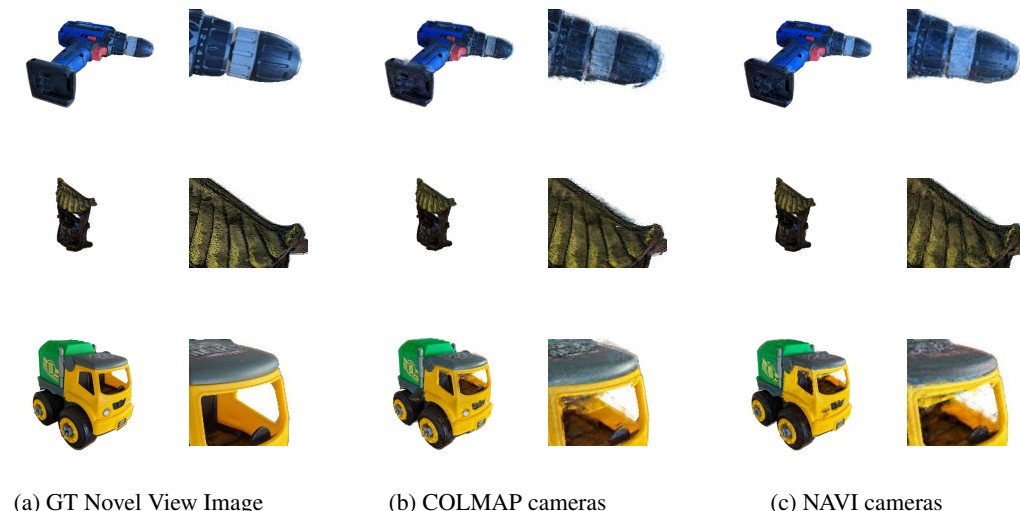

(a) GT Novel View Image      (b) COLMAP cameras      (c) NAVI cameras

Figure 13: **Novel view synthesis from multiview reconstructions**. We compare test examples from runs on selected scenes initialized with COLMAP reconstructed camera poses to NAVI GT pose initialization. For each configuration an InstantNGP [45] instance is optimized.

## B.2 COLMAP initialization details

Only 148 scenes of the 369 multiview scenes of NAVI could be completely reconstructed using COLMAP [52] with default parameters that are usually used in the context of NeRF [43] reconstructions. 155 scenes could be partially reconstructed with an average of 69% (± 34%) of the views registered. For the multiview collections included in the evaluation we randomly selected 5 scenes with more than 10 images for each of the objects. On those scenes COLMAP [52] successfully registered 73% (± 32%) of the views, on average. We only evaluate on the views of the validation set that were successfully registered by COLMAP, which gives the experiments initialized from COLMAP cameras an advantage.

## B.3 Visual results

Figure 13 illustrates qualitative results for novel view synthesis from multiview reconstruction using InstantNGP [45]. We compare the camera poses obtained by COLMAP with the ground-truth camera poses of NAVI. We observe some artifacts like the oversmoothed and noisy contours, especially on the COLMAP variant. These artifacts can be attributed to slightly offset camera poses, as well as the relatively small number of images in certain scenes for NeRF-like methods. We add a small amount of distortion loss as proposed by [4] to reduce the risk of floater artifacts. Additional regularization might be beneficial for some scenes to further improve results.

## C   3D from in-the-wild image collections

### C.1   Details of evaluated techniques

In the following, we briefly summarize the different techniques we analyzed for 3D from in-the-wild image collections on NAVI. For more details, please refer to the respective paper.

- **NeROIC [31]** proposes a multi-stage approach to reconstruct geometry and material properties from online image collections of objects. Camera poses are initialized with a COLMAP-based pipeline and fine-tuned during the first reconstruction stage which is followed by a normal extraction stage to estimate high-quality surface normals. Finally, material properties and illumination are estimated to enable relighting in addition to novel view synthesis.

- **NeRS [73]** introduces Neural Reflectance Surfaces that constrain reconstructions using a surface-based representation. Starting from manually annotated rough initial poses and a template mesh the objects are decomposed into a surface mesh, illumination and surface reflectivity as albedo and shininess. We define the dimensions of an initial cuboid that approximates the object's bounding box for each scene as suggested by [73].

- **SAMURAI [8]** enables reconstruction of a NeRF representation and decomposes appearance into illumination and a physically based BRDF based on a differentiable renderer for pre-integrated lighting [7]. Camera poses are initialized as quadrants from manual annotation and then refined using a multiplex and a coarse-to-fine scheme. We obtain the initial directions from the GT poses and use them to initialize both NeRS and SAMURAI.

- **GNeRF [41]** employs an adversarial approach and a pre-trained inversion network for camera pose and shape optimization from completely unknown cameras. GNeRF is the only method presented here that does not account for the different lighting settings which is also reflected in worse performance on more challenging scenes.

### C.2   Additional results

**Error distribution:** Figure 14b and 14d visualize the combined mean scores of PSNR and SSIM for the view synthesis task from SAMURAI and NeRS for all in-the-wild scenes. We observe that the difficulty of the NAVI scenes varies, as indicated by the fluctuating scores. By analyzing the results, we notice that the most challenging scenes across methods are the ones featuring symmetric objects, such as water guns and a hand drill. Our intuition is that symmetric objects of complicated shapes pose an additional challenge for these methods.

Figure 14a and 14c show the normalized view error and camera pose error over the views in one example scene. It is a common artifact that some cameras still show a high error after the optimization, probably being trapped in a local minimum. Interestingly, both NeRS and SAMURAI have a similar camera pose error distribution despite the differences regarding their 3d representation and camera optimization strategy. It's also important to note that large errors of individual cameras do not necessarily result in a failed shape reconstruction as long as there are enough correctly aligned views.

**View synthesis:** Figure 15 presents additional view synthesis results from NeROIC [31], NeRS [73] and SAMURAI [8] for selected test views. NeROIC is tailored towards high quality view rendering results which can be achieved with good initial poses. When initialized further away from the GT poses the lack of additional camera regularization leads to failing reconstructions. NeRS is able to robustly reconstruct all objects in most settings but shows limited quality for more complex shapes. Poses are generally less precise and scaled differently compared to the ground truth. SAMURAI is tuned towards large camera updates in the beginning of the training and therefore is able to reconstruct even from large initial pose offsets. Rendering quality is limited by the used illumination model. Results often show some smoothness, also a result of small pose errors that can not be compensated by the neural representation due to the explicit rendering step.

**Direct 3D Shape evaluation**

NAVI enables directly evaluating 3D shape reconstruction by using the provided 3D scans that are aligned on the images.

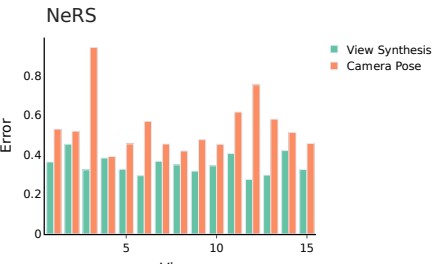

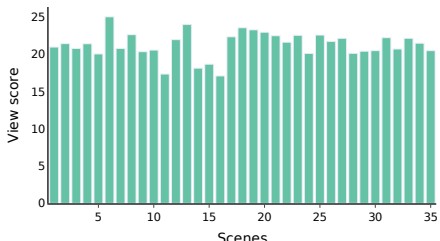

(a) Camera and view rendering errors over the test views for NeRS [73]

(b) Mean view rendering score over all in-the-wild-scenes averaged from SAMURAI and NeRS results.

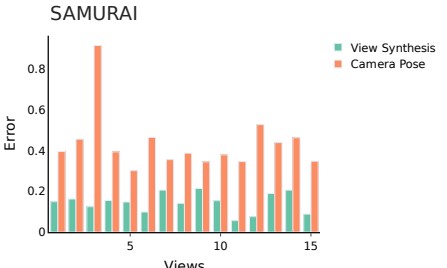

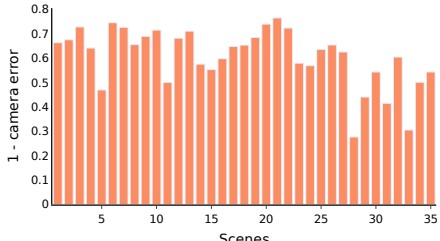

(c) Camera and view rendering errors over the test views for SAMURAI [8]

(d) Mean inverse camera error over all in-the-wild scenes averaged from SAMURAI and NeRS results.

Figure 14: **Error distribution over scenes and views**. Left we show the distribution of normalized view errors (inverse PSNR and SSIM) and camera pose errors (translation and rotation) over the test views of the "Keywest Showpiece" in-the-wild scene for two reconstruction methods, NeRS and SAMURAI. Lower values are better. On the right the normalized view score (SSIM and PSNR) and the normalized inverse camera error (translation and rotation) are given for all in-the-wild scenes in NAVI. Higher values are better. We report the average of SAMURAI and NeRS results.

| Method | Initialization | Mean Distance↓ | IoU↑ |
|---|---|---|---|
| NeRS [73] | Quadrants | 5.86 | 45.1% |
| SAMURAI [8] | Quadrants | 3.65 | 63.7% |

Table 8: **Shape evaluation from in-the-wild images.** The scanned 3D shapes provided by NAVI enable the evaluation of the reconstructed shape independently of the camera poses and rendering quality. We compare the point cloud extracted on the surface of the predicted mesh to points sampled on the ground truth mesh by using the Chamfer distance. We also measure Intersection-over-union (IoU) on fixed-resolution occupancy grid generated from the predicted and GT meshes.

For this experiment we use NeRS that generates a mesh as part of its optimization and SAMURAI that provides a mesh extraction pipeline. Generally, it is possible to generate point clouds from a NeRF representation which could also be compared to the GT mesh.

The reconstruction output of these methods is arbitrarily transformed by a rigid transformation due to the optimization setup. To align the predicted mesh with the GT mesh we first adjust its scale. We then perform fast global registration [75] on downsampled point clouds followed by refinement via point-plane ICP [50]. Figure 16 shows examples of the mesh alignment process.

For evaluation we use the mean Chamfer distance of the two set of vertices, and the 3D intersection over union (IoU) of the voxelized meshes. Results are presented in Table 8.

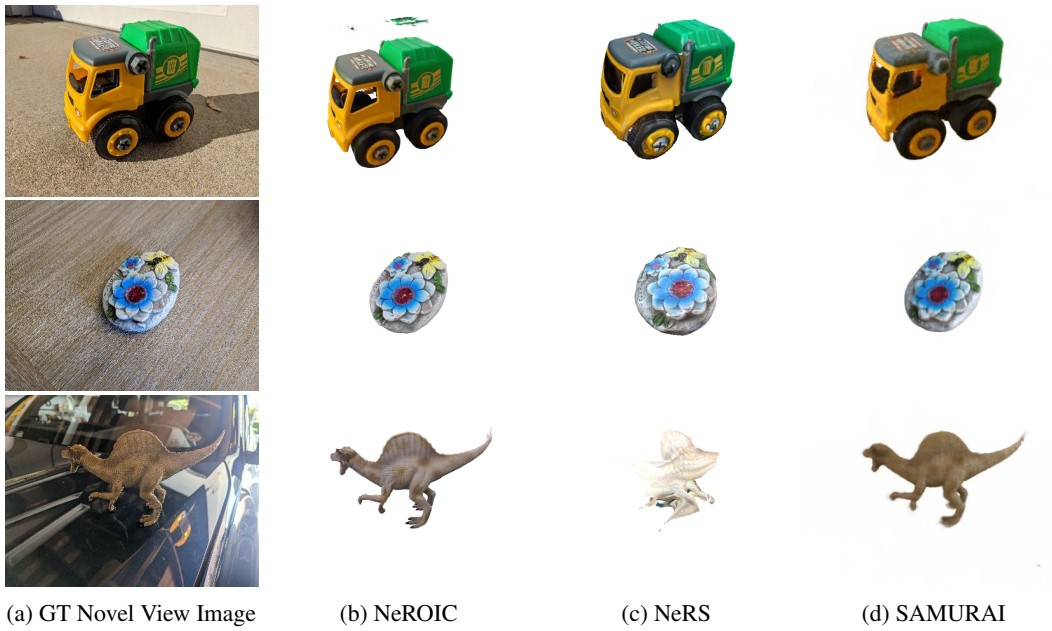

| (a) GT Novel View Image | (b) NeROIC | (c) NeRS | (d) SAMURAI |

Figure 15: **Novel view synthesis with in-the-wild 3D reconstruction**. Sample novel view synthesis results of different techniques for 3D reconstruction from in-the-wild image collections.

We observe that the relative reconstruction quality in terms of the shape metrics corresponds to the novel view synthesis results. Looking at the aligned meshes we can derive further insights. NeRS, which is initialized with a cuboid, fails to reconstruct objects with shapes that are very different from its initialization (see dinosaur in Figure 16a for NeRS vs. Figure 16c for SAMURAI). On the other hand, for objects with shapes closer to a cuboid NeRS tends to predict the bulk of the object more accurately (see less pink regions in Figure 16b), while SAMURAI is able to add finer details that are missing from the NeRS reconstruction (see Figure 16d).

## D  Correspondence estimation

### D.1  Details of experimented techniques

In the following, we briefly summarize the different correspondence techniques we benchmarked with NAVI. See the respective papers for more details.

- **SIFT + MNN/NN-Ratio [37]**. We use SIFT local features with heuristics-based matchers to represent the traditional baseline used in many Structure-from-Motion pipelines. Specifically, we use two popular variants of nearest neighbor search: mutual nearest neighbor and Lowe's ratio test.

- **SuperPoint + MNN/NN-Ratio [17]**. We replace traditional SIFT feature extraction with a learned detect/describe method, SuperPoint. We use traditional matchers to predict correspondences and rely on the improved descriptiveness of SuperPoint's features.

- **SuperPoint + SuperGlue [51]**. We replace traditional matchers with the popular SuperGlue sparse learnable feature matcher, which relies on a graph neural network and attention modules to predict correspondences from the input keypoint set. SuperGlue is trained on multiview image pairs from outdoor scenes, and we perform no additional object-centric finetuning.

- **LoFTR [57]**. Dense learnable matchers are often used to overcome repeatability issues in sparse local feature detection and matching. LoFTR relies on a coarse-to-fine transformer to propose a wider set of correspondences across entire images. We use the Kornia [49] implementation of LoFTR, which is also pretrained on outdoor scene pairs.

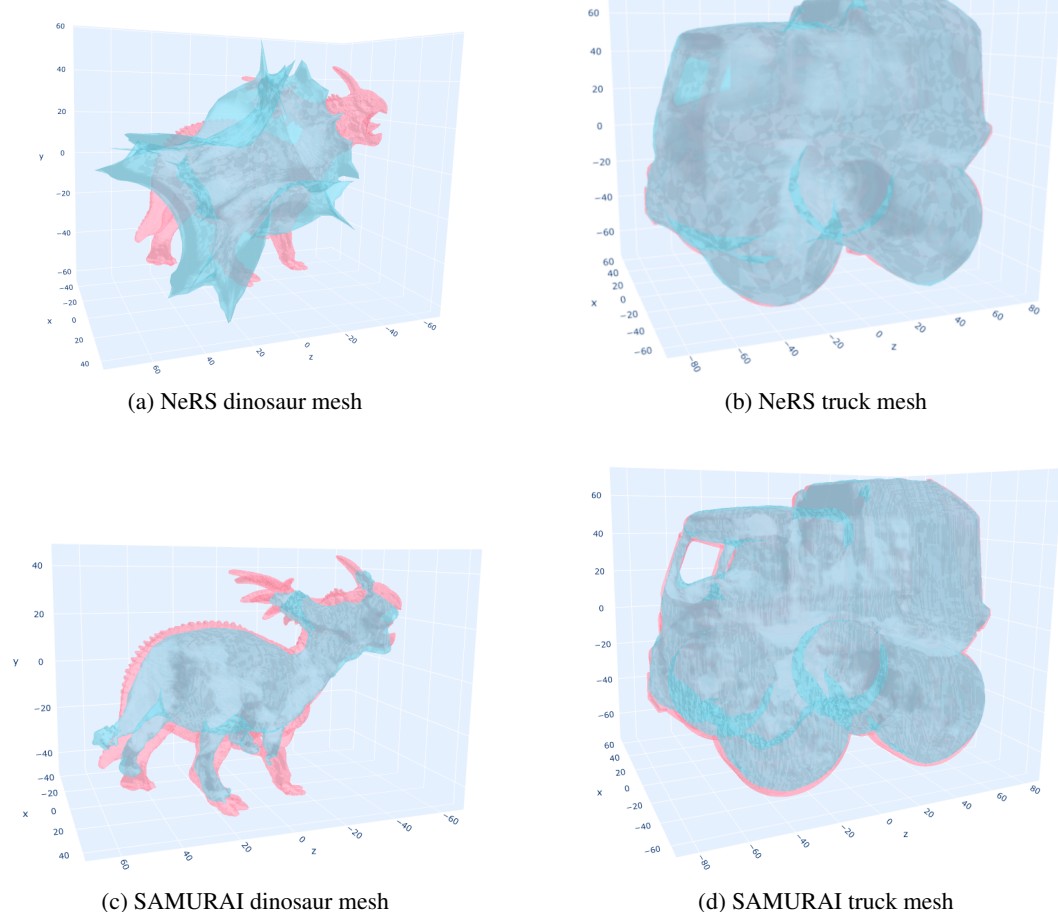

| (a) NeRS dinosaur mesh | (b) NeRS truck mesh |
| (c) SAMURAI dinosaur mesh | (d) SAMURAI truck mesh |

Figure 16: **Comparison of predicted and GT mesh after alignment**. The reconstructed mesh (blue) is aligned to the GT mesh (pink) and overlaid with a 50% blend. We show two methods: NeRS (top) and SAMURAI (bottom).

## D.2 Additional results

**Dense recall at varying thresholds**. Fig. 17 presents Dense-Recall@N results for the same set of correspondence estimation techniques, but with varying pixel radius'. We sweep radius values between 5 and 50 pixels and show that methods vary in dense recall performance at different threshold values. LoFTR and SuperPoint+SuperGlue outperform most methods in low-radius scenarios, but traditional matchers (SuperPoint+MNN and SuperPoint+NN-Ratio) see stronger performance for higher pixel radius'.

**Additional visual results**. Fig. 18 presents additional qualitative results for three correspondence estimation techniques. We believe these visualizations show that popular correspondence estimation techniques still have significant headroom for the task of fine-grained, object-centric correspondence estimation.

## E 3D from a Single Image

**Problem setting**. Given a single RGB image of an object, the aim is to reconstruct the 3D shape and optionally the 3D pose of the object depicted in it. Shapes are commonly represented as occupancy grids [47, 5, 12, 23, 63, 66, 67], point clouds [21, 38], or implicitly [46, 42, 11]. Poses are predicted relative to the camera, commonly as an explicit transformation [22, 32], or as part of the scene volume [47]. Single image 3D reconstruction is a highly ambiguous and under-constrained vision problem as the techniques have to reason about the complete 3D shape of the object from a single 2D projection of that object in a single environment.

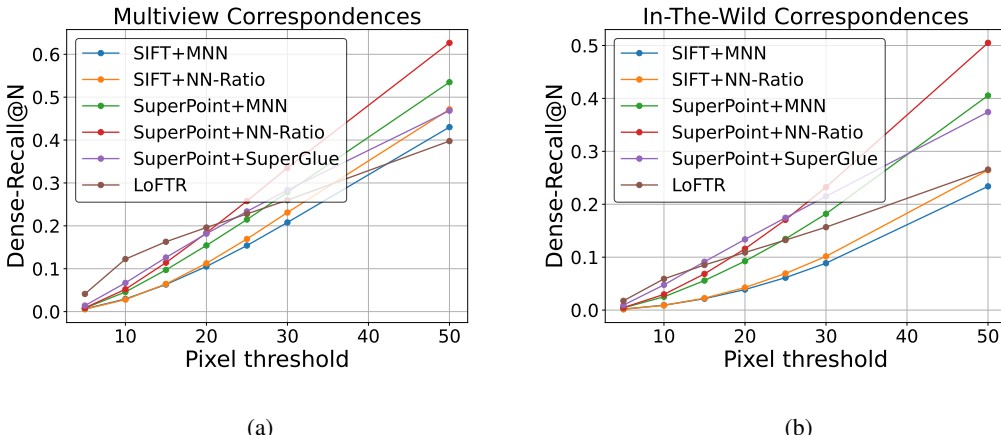

(a)                                                                                          (b)

Figure 17: **Dense-Recall@N,** where the pixel radius $N$ ranges from 5-50 pixels. Plots are provided for both the *multiview* and *in-the-wild* sets.

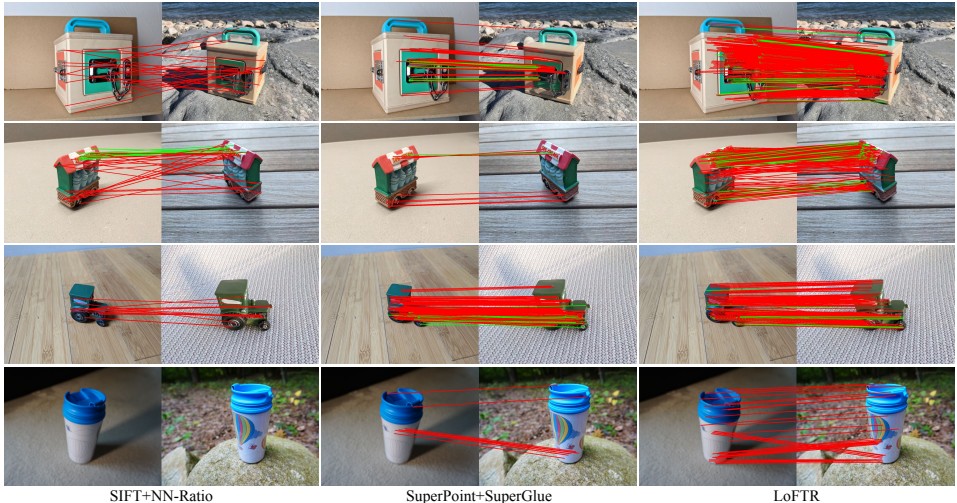

SIFT+NN-Ratio                          SuperPoint+SuperGlue                          LoFTR

Figure 18: **Sample correspondence results** of different techniques where the correct (within 3 pixels) and incorrect matches are shown in green and red respectively.

**The distinctiveness of NAVI**. Commonly used real-world datasets for single image 3D reconstruction such as Pix3D [58] and Pascal3D+ [65] are category-specific with objects of some common categories such as chairs, cars etc. As a result, simple recognition based 3D model retrieval techniques can already perform well on such class-specific datasets [59]. In contrast, NAVI objects are category-agnostic and provide a unique opportunity to evaluate the 3D geometric understanding capabilities of the techniques. Another key issue with the most existing real-world datasets is that the 3D shapes are only approximate (either nearest CAD models or reconstructed using SfM), whereas NAVI provides near-perfect 3D shape GT and alignments allowing for more accurate evaluations of 3D reconstructions.

**NAVI dataset and metrics**. In the experiments, we use all NAVI images, from both multiview and in-the-wild collection types. We split the images into train and test using three strategies: randomly ($S_i$), randomly along the object they depict ($S_o$), and along objects and environments ($S_b$). There are no images in common between the test and train splits in all three strategies. In addition, the set of object instances depicted on the images in $S_o$ and $S_b$ is disjoint between the train and test splits. Finally, the backgrounds against which objects are photographed are dissimilar between the train and test splits of $S_b$. We use the official NAVI splits for $S_i$ and $S_o$ and we rely on capture location to split the dataset for $S_b$. $S_i$ and $S_o$ follow the practice of [47, 22] for measuring performance in datasets with real images. Splitting along images in $S_i$ allows the model to learn about the geometry of all objects in the dataset and to apply this knowledge to images it hasn't seen in the test split. $S_o$ presents

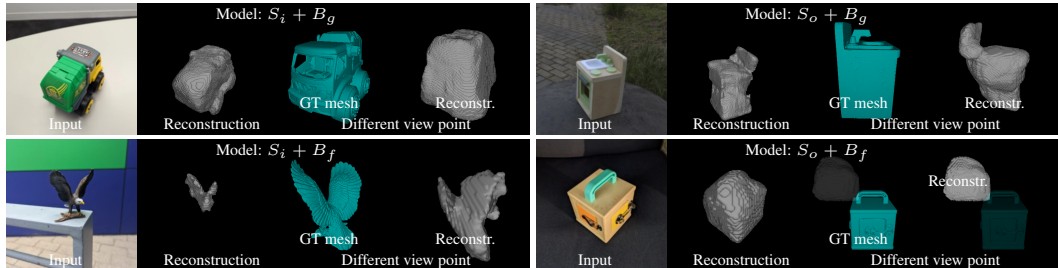

Figure 19: **Sample single image 3D reconstructions using CoreNet [47]**. In all cases, the reconstructed geometry aligns well with the input image. When splitting along objects ($S_o$), reconstructions contain errors in unobserved parts. In addition, CoReNet cannot resolve the depth/scale ambiguity for $S_o + B_f$ and it reconstructs objects at a wrong depth. Both are evident when viewing reconstruction from a different view point.

|  | images ($S_i$) | objects ($S_o$) | backgrounds ($S_b$) |
|---|---|---|---|
| GT pose provided ($B_g$) | 66.6% | 46.3% | 46.2% |
| Fixed grid ($B_f$) | 49.7% | 13.1% | – |

Table 9: **IoU performance of CoReNet on the NAVI dataset, under different settings**. Columns indicate the way data is split into train and test, rows – whether the model has access to the ground-truth pose at test time.

a harder scenario, as the model has to reconstruct unseen geometry on unseen images. $S_b$ poses an additional challenge, as the model can no longer rely on a familiar background on the test set.

**Experiment setup**. For our preliminary experiments, we use CoReNet [47] as a representative single-object reconstruction method. CoReNet predicts volumetric binary occupancy on a regular grid inside a given 3D box in front of the camera. The geometry and the pose of the object can be extracted from the grid using Marching Cubes [33]. We establish baselines by training and evaluating CoReNet on the NAVI dataset. We evaluate CoReNet in two scenarios: 1. We provide CoReNet with access to the object's GT pose at test time ($B_g$), using the mechanism described in the paper [47] for the Pix3D dataset and; 2. We ask CoReNet to reconstruct occupancy inside a fixed $3m \times 3m \times 3m$ bounding box, placed $1m$ in front of the camera ($B_f$). Scenario $B_g$ is essentially equivalent to shape prediction. Scenario $B_f$ combines shape and pose prediction. It is much harder than $B_g$, since the model has to resolve the depth/scale ambiguity from a single image only. It is more feasible in NAVI, as NAVI has a single prominent object in each image and the pose is given in metric space.

### E.1 Analysis

We train CoReNet models on 5 combinations of split strategy and pose prediction. In all cases, we start from a model pre-trained on synthetic data ($h_7$ from the CoReNet paper), and we train for 15 epochs. To evaluate, we measure intersection-over-union (IoU) between the predicted and ground truth occupancy grids. Table 9 summarizes the results, Figure 19 shows sample reconstructions.

**Analysis across different splits**. Models trained on $S_i$ outperform those trained on $S_o$ by a large margin (+20.3% for $B_g$, +36.6% for $B_f$). As expected, splitting randomly along images allows the model to learn about object geometry and to apply this to the similar objects in the test set. This is also confirmed visually (Figure 19). Reconstructed objects re-project correctly over the input images in all scenarios, but they contain large errors in unobserved regions for $S_o$. The difference between models trained on $S_o$ and $S_b$ is negligible, indicating that learning about geometry is more important than background.

**Analysis with and without pose prediction**. Comparing models with access to the ground-truth pose ($B_g$) to those without ($B_f$), shows that performance falls modestly for models trained on $S_i$ (from 66.6% to 49.7%) and significantly for models trained on $S_o$ (from 46.3% to 13.1%). Learning about model geometry and most importantly about object size becomes essential for $B_f$, as the model has no other means to resolve the depth/scale ambiguity. Visually, reconstructed objects re-project correctly over the input images, but looking from below reveals that they are smaller/larger than the ground truth and they are reconstructed at the wrong depth (Figure 19).

