# OpenReview forum: "NAVI: Category-Agnostic Image Collections with High-Quality 3D Shape and Pose Annotations"
_NeurIPS.cc/2023/Track/Datasets_and_Benchmarks — NeurIPS 2023 Datasets and Benchmarks Poster_

### Official Review · Reviewer_Ymgi · 2023-07-20
**A dataset for evaluating SfM methods "in the wild"**

**Rating:** 7
**Confidence:** 4

**Strengths:**

The authors propose the first dataset for in the wild multiview stereo evaluation.

The dataset allows indirectly to test a variety of related tasks.

**Additional Feedback:**

Nothing to add.

**Clarity:**

The structure of the paper is a bit confusing. Section 3, 4 and 5 might be better assembled in a single section "experiments".

**Correctness:**

I found not statements that is incorrect, albeit the way the dataset is constructed could be made more explicit in section 2.

**Documentation:**

The dataset is made available through a website. I would recommend the authors make the license of the dataset visible on the website itself.

**Limitations:**

The authors have discussed the limitations of their work.

**Opportunities For Improvement:**

The dataset design part of the paper is really short, which makes it hard to evaluate the actual quality of the pipeline. One of the major question I have is how the intrinsic parameters of the cameras are accounted for?

For example I am not sure I agree with the sentence "Relative camera poses are an implicit output of alignment, as all objects were posed with respect to their canonical pose. " on line 121. How is the focal length calibrated? I cannot assume that the intrinsic camera parameters are perfectly known for all images in the wild, or are they?

Overall, the data curating part of the paper (section 2) should be extended to present more details on the pipeline itself, while the next sections might be condensed into a single results sections, or moved to the supplementary material.

**Relation To Prior Work:**

Yes.

**Summary And Contributions:**

The authors propose a new dataset for evaluation of Multiview Stereo techniques on in the wild images.
The dataset is shortly presented, then multiple experiments are conducted on multiview stereo reconstruction and associated problems like correspondence finding or relative camera pose estimation.

---

> ### Author Response · Authors · 2023-08-22
> **Author Response**
>
> ---
> ### **The dataset design part is really short. The data curating part (section 2) should be extended.**
>
> Thanks for the suggestion to expand the dataset design section. We use much of the additional page in the revision to add sub-section 2.2 titled ‘Dataset analysis’ with paragraphs on
> ‘General statistics’, including Table 1 (moved from the supplementary).
> ‘Annotation quality analysis’,
> ‘Manual vs. semi-automatic alignment process’
> ‘On using self-captured vs. internet images’.
> We hope this provides enough information on dataset analysis and rationale for our annotation process.
>
>
> ---
> ### **How are the camera intrinsics accounted for?**
>
> Since we captured all the NAVI images ourselves, we know the device used to capture the images (all capturing devices are listed in Table A1 of supplementary). The focal length of the images was initialized from the ExIF metadata, which was available in our captures. The annotators have the option to modify the focal length from within the annotation plugin, in cases that the projections look wrong. In the majority of cases, the camera intrinsics from ExIF metadata is sufficient for obtaining high-quality 2D-3D alignments.
>
> Apart from that, our annotation tool handles cases where the camera intrinsics are completely unknown eg. when annotating on internet images (Ablation of Figure A4 in supplementary). In these cases, the annotators adjusted the focal length manually.
>
>
> ---
> ### **The structure of the paper is a bit confusing. Section 3, 4 and 5 might be better assembled in a single section ‘experiments’ section.**
>
> Since NAVI is a multi-facet dataset that can be used for different problem settings, we believe it is important to view the dataset from the lens of these different problem settings. A significant portion of these sections is about the NAVI relation to this problem setting and its uniqueness w.r.t. existing datasets for that particular problem. These are not just experimental sections. Experiments in these sections add more concrete evidence on why NAVI is a useful dataset for evaluations in these different problem settings, while also providing interesting insights on the existing techniques for these problems.
>
> ---
> ### **Make the license of the data visible on the project web page itself.**
>
> Thanks for the suggestion. We added the license below the teaser figure in the project [webpage](https://navidataset.github.io/)

---

> > ### Comment · Reviewer_Ymgi · 2023-08-24
> > **Following the authors' response**
> >
> > I would like to thanks the authors for promptly addressing the reviewers comments.
> >
> > I think the improvements proposed in the revised version significantly improve the quality of the proposed dataset. The new section 2.2 provides a lot of useful information. Especially, the estimation of the alignment quality and comparison with existing datasets gives a stronger argument in favor of the proposed dataset.
> >
> > The organization of the paper is still sub-optimal in my opinion, but at this point I am willing to admit that it comes down to preferences.
> >
> > From my point of view, the paper can be accepted. I have updated my rating accordingly. But I would recommend that the author add a comment on the obtention/estimation of intrinsic camera parameters in the final version of their paper.

---

> > > ### Author Response · Authors · 2023-08-24
> > > **Thanks for your response**
> > >
> > > Glad to see your positive response and thanks for updating the rating. We will expand L104-105 of the main paper to clarify the use of intrinsic parameters.

---

### Official Review · Reviewer_kSQG · 2023-07-20
**Great dataset quality; but requires exhaustive manual labor.**

**Rating:** 6
**Confidence:** 4
**Clarity:** It was easy to follow.

**Strengths:**

This work emphasizes the quality of scanned objects. The annotations given by the 2D-3D alignment are beneficial for systematic research progress on 3D reconstruction and correspondence estimation. In addition, the authors clearly elaborate experimental details.

**Additional Feedback:**

The authors should clearly state whether they will release the dataset generation code with the program they used for aligning 2D-3D.

Typo
L95 -> multiview captures offer the… → Multiview captures offer the (capitalize)

**Correctness:**

The methodology used for dataset generation seems clear and well-defined. Moreover, their evaluation sufficiently demonstrates the excellence of the proposed dataset.

**Documentation:**

Clearly stated.

**Ethics:**

I think there are no ethical problems regarding the proposed dataset.

**Limitations:**

The authors should acknowledge that their approach has limitations when it comes to generating a large number of instances with significant diversity. Due to the limited number of objects used for training, the network may struggle to produce a vast array of instances that encompass a wide variety of backgrounds and scenarios. It is essential to address this constraint honestly and consider potential future work or alternative approaches to overcome this limitation and achieve greater diversity in instance generation.

**Opportunities For Improvement:**

- The description for the alignment verification steps lack clarity. For example, terms like “slightly wrong” and “occluded by other objects” are too vague to reproduce the dataset generation accurately. If manual verification was necessary, the authors should state their criteria for labeling “incorrect alignment” and “occluded” instances. It would be beneficial if the authors could improve the description of their alignment verification step to provide more detailed and understandable information.
- Many steps of the data generation process are based on manual labor. This limits the extension of the proposed data to larger volume. Moreover, the construction process relies on expensive 3D scanners so the number of objects in NAVI is not sufficient.
- Based on the attached video in the supplementary material, the 2D↔3D alignment process appears to be prone to inaccuracies as annotators need to decide when the alignment is complete. To make the alignment process less exhaustive and subjective, a suggestion could be to employ a masked MSE loss and optimize the pose during test-time to minimize the masked MSE loss. This approach may lead to more reliable and consistent results.
- The limited number of objects used for training can result in the trained network learning correspondences primarily from these objects, without considering the diverse backgrounds they appear in. Consequently, this could lead to a bias in the network's ability to generalize well to various backgrounds, as it may not have been exposed to a wide range of background variations during training. If we were to train a network for correspondence from scratch in NAVI, it could become biased, leading to poor performance when transferred to other datasets. This indicates the importance of addressing the limitations and potential biases during the training process to ensure the network's effectiveness and generalizability across different datasets.
- They do not report failure cases of dataset generation step. I’m not sure what difficulty makes the failure cases.
- It would be be good to add the papers following (not necessary but just a suggestion for further improvement):
    - Yu, Xianggang, et al. "Mvimgnet: A large-scale dataset of multi-view images." *Proceedings of the IEEE/CVF Conference on Computer Vision and Pattern Recognition*. 2023.

**Relation To Prior Work:**

The relevance to the prior work is clearly stated. In the supplementary material Table 2, the authors should add CO3D and (optionally) MVImgNet (this paper is introduced in CVPR; after the NeurIPS submission)

**Summary And Contributions:**

This work proposes a new object image dataset, called NAVI, consists of category-agnotic image collections. The dataset includes high-quality 3D scans and near-perfect GT camera parameters obtained by 2D-3D alignment. Additionally, it contains the derivative annotations from the 2D-3D alignment, such as dense pixel correspondences, and depth, segmentation maps. The proposed dataset is beneficial for systematic research progress on 3D reconstruction and correspondence estimation. To demonstrate excellence of the proposed dataset, they conduct various experiments.

---

> ### Author Response · Authors · 2023-08-22
> **Author Response**
>
> ---
> ### **Criteria for labeling “incorrect alignment” and “occluded” instances**
>
> For the alignment/verification quality, we instructed the verifiers to be *very* strict. In fact, at the beginning of the project we verified many alignments ourselves, and chose the most accurate annotators (aligners) as the verifiers. The instructions are clear and simple: “If the object does not align perfectly, discard the alignment”. For a quantitative evaluation of how accurate the resulting alignments are, please refer to the general response paragraph on **2D-3D alignment quality and inter-annotator consistency**, which we also added to the main paper (lines-136-149). For occlusions, we also instructed the raters to be strict, and mark an object as “occluded” even if a very small portion of it is occluded (eg. a leaf occludes the object).
>
> ---
> ### **Manual alignment process limits the data annotation. Isn’t it better to use some silhouette or point-based loss functions to semi-automate the annotation process?**
>
> We agree that the proposed pipeline is not scalable. However, to achieve the high-quality annotations that we provide, we needed to sacrifice quantity over quality. In fact, we started with semi-automatic approaches like using keypoint correspondence annotations and silhouette based optimization. None of these provided the high-quality alignments we are aiming for in the NAVI dataset. For a detailed analysis, please refer to the  general response on **Annotation rationale**, where we illustrated that point correspondences based 2D-3D alignment (that is often used in existing datasets) has quality issues.
>
>
> ---
> ### **Number of objects in NAVI is not sufficient. Training a correspondence network on NAVI can have poor generalization to unseen objects.**
>
> See the above general response paragraph: **On the number of objects (size) in the dataset**
>
> We agree that the number of objects in NAVI are not enough for training a generalizable correspondence network. The main use of NAVI is for 3D reconstruction from image collections. For correspondence estimation, NAVI provides a unique *evaluation* dataset due to its availability of dense correspondence ground-truth not present in other real-world datasets. NAVI also contains correspondence data for the same object in diverse backgrounds and illuminations, which allows for evaluating object-centric matching from varying scenes. This serves as a challenging benchmark for evaluating matching techniques, since matchers must rely solely on object content for matching.
>
> ---
> ### **Failure cases of data generation step.**
>
> Failures in 2D-3D alignments usually happen for the following reasons:
> The object appears blurry in the image.
> Small parts of the objects are mis-aligned (annotator error).
> The object has slightly deformed from its initial shape.
> The focal length initialization is not good.
> Note that only 7% of the alignments fail our strict verification process (L115 main paper).
> For a quantitative evaluation, please refer to general response on **2D-3D alignment quality and inter-annotator consistency**.
>
> ---
> ### **Reference to MVImgNet.**
>
> Thanks for the suggestion. We added this reference, and we added MVImgNet in the comparisons to multiview datasets (Table 2, supplementary).
>
> ---
> ### **On releasing the annotation tool.**
>
> We agree that this annotation tool would be useful for the community. We will de-couple it from internal software and release it.
>
>
> ---
> ### **On ethical concerns.**
>
> We believe that the reviewer mistakenly flagged  the option  ‘1. Yes, there are significant ethical concerns’. We would be happy to address any concerns, nevertheless.

---

> > ### Author Response · Authors · 2023-08-27
> > **Friendly reminder**
> >
> > Dear reviewer kSQG, we addressed your concerns to the best of our abilities in our rebuttal, and the revised version of the paper. With the discussion period ending soon, we would like to kindly ask if there are any additional concerns or suggestions for our rebuttal.
> >
> > Thank you for your time.
> > The NAVI authors.

---

> > > ### Author Response · Authors · 2023-08-31
> > > **Last day of author-reviewer discussions**
> > >
> > > Dear reviewer kSQG,
> > >
> > > Thank you again for your constructive feedback, and for helping us improving our paper.
> > > Today is the last day of the author-reviewer discussion phase.
> > > Could you please let us know if our rebuttal and revised version of our paper addressed your concerns?

---

### Official Review · Reviewer_vLyG · 2023-07-21
**A Promising Dataset for Shape and Pose Estimation From Images**

**Rating:** 7
**Confidence:** 3

**Strengths:**

- Visual quality of alignment annotations.
- Supplemental provides exhaustive relevant detail
    -- Licensing information
    -- Public release of data and code (with tutorial)
    -- Nice project webpage
    -- Implementation details for experiments
    -- Experiment qualitative result comparison, additional experiment constructing 3D from a single image
- Dataset size and setup instructions make it easy for others to use.

**Additional Feedback:**

The authors propose a relevant and useful dataset, and demonstrate its utility for multiview object reconstruction, 3d shape/pose estimation from in-the-wild images, and pixel correspondence estimation. The data is extensively validated and appears to be of high quality. The paper is well written and also of high quality. My only concern is that the dataset is relatively small in terms of object number which will limit its generalizability and prevent it from being used to train most machine learning models. This reviewer would like to see a commitment to expanding the dataset over time and releasing the annotation tool if possible. However altogether the authors make several strong contributions. Therefore I recommend an accept.

**Clarity:**

This paper is well-written with a few minor exceptions. I recommend making an editing pass on the paper for the CR version.

**Correctness:**

Dataset seems to be constructed in a sound manner. available data is of good quality, and shown visualizations are representative of the existing data and annotations.

**Documentation:**

There is extensive documentation concerning how to access and use the data. Licensing is discussed. Intended use case discussion is limited to the experimental sections. Unfortunately, no hosting/maintenance plan. Experiments are reproducible with the instructions, implementation details, and code available.

**Ethics:**

I have no ethical concerns with this work.

**Limitations:**

As previously mentioned, dataset size is likely a limiting factor for broader deep learning applications, e.g., training shape reconstruction from images.

**Opportunities For Improvement:**

- Releasing the annotation alignment tool will surely benefit other researchers and should be done soon, if possible.
- Would prefer to see more of a breakdown of dataset composition, i.e. how often each type of device was used, etc.
- Lacking specific detail about data collection - were object scans cleaned or processed in any way aside from origin centering? what 12 devices were used for data collection? what conditions were sought after for each type of environment? how was inter-annotator and inter-data-collector consistency assured?
- As acknowledged by the authors, the dataset size is not very large, especially for training in broader (e.g. shape reconstruction from images) machine learning applications. Given the size of the dataset, I would like to see some commitment to expanding the dataset over time, especially given the institutional resources available. Right now the dataset is rivaled in object count by relatively old datasets, e.g., YCB. Though the authors show their approach may be used for fine-tuning, image/scanned-object paired datasets will likely become increasingly important to shape learning tasks in the future due to the relative complexity of scanned objects with respect to CAD objects, and creating large datasets of scanned objects (as Google has done before with GSO) would be extremely helpful to the community.

**Relation To Prior Work:**

In every benchmark context, NAVI is distinguished from prior work in some way.

**Summary And Contributions:**

The paper proposes the following contributions:
1. Dataset including collections of 10,515 images of objects from different, known cameras. collections are (majority) from somewhat controlled (“casual”) settings, and (minority) from in-the-wild settings which are much more free.
2. Each image is paired with high-quality and seemingly trustworthy 2d-3d camera pose annotations, segmentation, monocular depth
3. Each collection of images is supplemented with dense correspondence points between image pairs.
4. Dataset usefulness demonstrated through 3 experiments: multiview object reconstruction, 3d shape/pose estimation from in-the-wild images, pixel correspondence estimation.

---

> ### Author Response · Authors · 2023-08-22
> **Author Response**
>
> ---
> ### **On releasing the annotation tool.**
>
> We agree that this annotation tool would be useful for the community. We will de-couple it from internal software and release it.
>
> ---
> ### **Breakdown of data composition i.e. how often each type of device is used.**
>
> Please refer to the general response paragraph: **Overview of dataset statistics**. We also added Table 1 (in supplementary) of all camera devices used in NAVI.
>
>
> ---
> ### **Were the object scans cleaned or processed aside from origin centering?**
>
> The scans are first manually curated and cleaned using the scanner software. We then bring them to a canonical pose where the Y+ axis is up, and the Z- axis is front (like in the ShapeNet dataset). We center them at origin, but we do not normalize them, in order to retain their real-world metric dimensions. Please refer to [Figure A2](https://drive.google.com/file/d/1yqG242PU4r0KBak2j4p5EH4JKnsh9NcG/view?usp=drive_link) in the supplementary for the quality of 3D scans.
>
> ---
> ### **How was the inter-annotator and inter-data-collector consistency assured?**
>
> For the inter-annotator consistency, we trained all the raters using the same protocol and we were quite stringent to reject an annotation even if the 3D model is slightly mis-aligned. See the above general response paragraph on **2D-3D alignment quality and inter-annotator consistency**, where we quantitatively evaluated the resulting alignments using inter-annotator consistency as the metric.
> For the inter-data-collector consistency, we only have rough guidelines as we want to mimic the diversity in the real-world casual captures.
>
> ---
> ### **Dataset is relatively small in terms of the number of objects.**
>
> See the above general response paragraph: **On the number of objects (size) in the dataset**
>
> ---
> ### **On plans to expand the dataset over time.**
>
> We showed that the current NAVI dataset is already useful for evaluation of 3D reconstruction techniques as this enables more robust evaluations of poses, correspondences etc. compared to prior works. Nevertheless, we are committed to expanding the dataset over time, see the above **Annotation Rationale** response paragraph: **On expanding the dataset**.
>
> ---
> ### **Hosting/Maintenance plans**
>
> Hosting / Maintenance plans are included in section 1.5 in the supplementary material:
> “The  data  is  hosted  on  Google  Cloud  by Google Research. The authors will maintain and update the dataset.”

---

> > ### Comment · Reviewer_vLyG · 2023-08-25
> >
> > I appreciate the prompt and detailed response on the part of the authors.
> >
> > The inclusion of easily interpretable dataset statistics is helpful for making the dataset stand out at a glance for potential researchers. The authors explanation of object processing and consistency checking is well-founded. I appreciate that the authors have made a commitment to release their annotation tool, and that the authors plan to continue to add to the dataset. I believe that more high quality data in this field is extremely important. However, I think that the size of the dataset is still somewhat underwhelming. Though the size may be competitive compared to existing collections used with neural radiance fields, the richness of the dataset lends itself to so many other applications that I would have like to see more objects from the start, and post-publication expansion of the dataset is likely to be slow. Therefore, I think my rating is still appropriate.

---

> > > ### Author Response · Authors · 2023-08-26
> > > **Thanks for your response**
> > >
> > > We are glad that reviewer vLyG is satisfied with the rebuttal. We will make this dataset larger over time, as stated in General Response, “On expanding the dataset”.

---

### Official Review · Reviewer_eykt · 2023-07-22
**A valuable dataset research direction but seems not sufficient data and experimental settings**

**Rating:** 5
**Confidence:** 4
**Correctness:** Should be mostly correct.

**Strengths:**

1. In contrast to previous datasets, the proposed dataset offers a remarkable collection of accurately captured multiview 2D images and their corresponding 3D scans, many obtained under real-world, in-the-wild conditions.

2. The creation of this dataset involved many efforts to achieve near-perfect 2D-3D alignments, ensuring high-quality and precise data.

**Additional Feedback:**

1. As only 36 3D objects have been utilized, it is suggested to show all the 3D objects.

2. Why consider "extending the dataset to include videos" instead of adding more 3D objects?


**Clarity:**

Clearly presented, but the insight behind is not clearly justified.
It is also strange as the statistics of this dataset is not provided in the main paper.

**Documentation:**

yes.

**Ethics:**

no.

**Limitations:**

Yes.

**Opportunities For Improvement:**

1. It is important to note that this dataset comprises only 36 3D objects, which significantly limits its potential for various downstream tasks. The limited size might pose challenges in training networks with sufficient generalizability, making it more suitable primarily for evaluation purposes.  I would suggest utilizing more 3D objects, which is very important.

2. I strongly recommend including detailed comparisons between the proposed NAVI dataset and existing datasets. This will offer valuable insights into the unique characteristics and advantages of NAVI over its counterparts. Additionally, it is crucial to emphasize the necessity of this dataset for related downstream tasks, as its specific applications might not be immediately apparent.

3. I am curious about the possibility of showcasing how the utilization of the NAVI dataset could facilitate real-world applications.  If the authors could clearly show the advantages of using the NAVI dataset, the significance of this dataset could become clearer.

**Relation To Prior Work:**

This work discusses the differences, but it does not provide detailed comparisons.

**Summary And Contributions:**

This work presents the NAVI dataset, a category-agnostic 3D object dataset with in-the-wild multi-view image collections, high-quality per-image 2D-3D alignments, and near-perfect GT camera parameters.
This dataset provides 36 3D objects and 10515 images with 2D-3D alignments, captured in 267 unique scenes.
The authors demonstrate this dataset could possibly benefit 3D reconstruction and correspondence estimation.

---

> ### Author Response · Authors · 2023-08-22
> **Author Response**
>
> ---
> ### **Dataset only comprises 36 objects which limits its use for downstream tasks.**
>
> Please refer to the above General Response paragraph: **On the number of objects (size) in the dataset**
>
> ---
> ### **Detailed comparisons between NAVI and existing datasets**
>
> Please refer to the above General Response paragraph: **Comparison to other related datasets**
>
> ---
> ### **Advantages of using NAVI dataset and its use in real-world applications**
>
> NAVI was extensively used for real-world applications in this paper. In fact, NAVI enables evaluations on real data that were not possible before. Some of the highlights are:
> In Sections 3 and 4, where NAVI was used to train for 3D reconstruction from image collections. NAVI provides better camera poses than COLMAP in its multiview images (ground-truth quality), and most importantly, provides the utilities needed for direct 3D shape evaluation of existing methods (e.g., [Figure A8](https://drive.google.com/file/d/1HxF-WNF_IqVQF3eJrUJsGL-tg_s99Rcy/view?usp=sharing) in supplementary).
> In Section 5, where NAVI enables evaluating for dense correspondence estimation.
>
> In case the reviewer refers to *training* on NAVI for downstream tasks: As we  highlight in the general response “**On the number of objects (size) in the dataset**”, this is not the intended usage of NAVI, and there are plenty of datasets for that. Our main contribution is that we enable  robust evaluations on image collections that are essential for progress on 3D reconstruction techniques.
>
> ---
> ### **Show all the 3D objects**
>
> We visualize all the 3D shapes in NAVI, sample images, and their 2D-3D alignments in these [Slides](https://docs.google.com/presentation/d/1LCWUBQHs3oGi1bwCQjgLm8-etV9Y8ldPSpFN0BLoi6o/edit?usp=sharing), which we linked to our [NAVI git repo](https://github.com/google/navi).
>
>
> ---
> ### **Why consider extending the data to include videos instead of more 3D objects?**
>
> We are actually working on both: more objects, and annotating videos. Video annotations enable the use of NAVI on more video related 3D reconstruction tasks (consecutive frames, blurrier captures than images). Adding more 3D objects can help with more robust evaluations. We believe that 36 objects is already a good number for robust evaluation of 3D reconstruction techniques that optimize *within* image collections. We are committed to expanding the dataset over time, see the above **Annotation Rationale** response paragraph: **On expanding the dataset**.

---

> > ### Author Response · Authors · 2023-08-27
> > **Friendly reminder**
> >
> > Dear reviewer eykt, we addressed your concerns to the best of our abilities in our rebuttal, and the revised version of the paper. With the discussion period ending soon, we would like to kindly ask if there are any additional concerns or suggestions for our rebuttal.
> >
> > Thank you for your time.
> > The NAVI authors.

---

> > > ### Comment · Reviewer_eykt · 2023-08-27
> > >
> > > Thanks for the rebuttal and the revision.  The revised version has improved a lot.
> > > I also checked the comments of the other reviewers.
> > > However, I still have the following concerns:
> > >
> > > (1) limited size:
> > > Although the authors argue that the proposed dataset is larger than a few multiview datasets, and the other large multiview datasets, such as CO3D, do not provide high-quality camera poses, the size of the current NAVI dataset is truly very limited.
> > > It would not change the current situation, as we still need to train or tune our method by using existing datasets, although NAVI could be used for evaluation.
> > > In short, although it is better than a few existing datasets, it might not be good enough.
> > >
> > > (2) applications:
> > > The authors highlight two main applications:
> > > a. 3D reconstruction from image collections: I agree that NAVI might be able to provide better camera poses than COLMAP in its multiview images(ground-truth quality), facilitating nerf-related research works for reconstructing the "foreground objects" under different environment conditions.
> > > b. Correspondence and relative pose estimation: Actually, the listed methods, such as SuperGlue and LoFTR are not specifically designed for objects, and they actually prefer to use the images for diverse indoor/outdoor scenes.
> > > Hence, I am worried that the NAVI dataset seems to be not very interesting for this application.
> > > Therefore, I asked the authors if the proposed NAVI dataset could facilitate any more real-world applications, but the current answer did not fully resolve my concern.
> > >
> > > (3) all the 3D objects：
> > > Why are certain 3D objects textured meshes while others are not?

---

> > > > ### Author Response · Authors · 2023-08-28
> > > > **Further clarifications**
> > > >
> > > > We are glad that you find the new version of the paper improved. Thank you for your further clarification questions and sharing your concerns. Here are our responses:
> > > >
> > > > - **On limited dataset size**: In NAVI, our aim is to provide *new and complimentary* aspects of high-quality 3D, and 2D-3D alignments which are missing in existing datasets. For tasks like multi-view 3D reconstruction (joint shape and pose estimation), the quality of ground truth is more important than the quantity, as training and evaluations are performed *within* image collections. Our dataset enables direct shape and pose evaluation for these tasks, because of the high-quality alignments. For other tasks such as correspondence estimation, our dataset enables *new, complimentary evaluation* of dense geometric correspondences, which is not possible for other datasets that don’t provide high-quality alignments.
> > > >
> > > > - **NAVI for correspondence estimation**: We agree that existing works such as SuperGlue and LoFTR are mostly designed for scenes but not for objects. We argue that this is partly due to the non-availability of high-quality object centric datasets. NAVI partly fills this gap by providing a useful evaluation dataset for object-centric geometric correspondences. Note that even standard NeRF pipelines rely on correspondence estimation for inferring the camera poses, typically on object-centric datasets.
> > > >
> > > > - **More applications**: NAVI is useful for evaluating other object-centric geometry tasks such as single-image 3D shape estimation, depth estimation, surface normal estimation, and foreground segmentation. We did not highlight these applications in the paper as there exists several other datasets for these tasks and NAVI is mainly useful for evaluation purposes. Instead, we focus on the applications where NAVI unlocks new and complementary advantages: 3D from image collections, and correspondence estimation.
> > > >
> > > > - **Why do some objects have texture maps while others do not?**: Two different authors scanned the objects with two different scanners: 1. Einscan-SP and 2. Einscan-Pro-HD. Both have very similar precision in terms of geometry (0.05mm). Scanning with the first one requires spray-painting the object and thus texture maps do not correspond to the object texture. The second scanner does not require spray paint and we also released the texture maps for those scans (although we did not use them in our experiments).

---

> > > > > ### Author Response · Authors · 2023-08-31
> > > > > **Last day of author-reviewer discussion.**
> > > > >
> > > > > Dear reviewer eykt,
> > > > >
> > > > > Thank you again for your constructive feedback, and for helping us improving our paper.
> > > > > Today is the last day of the author-reviewer discussion phase. Can you please let us know if our responses addressed your concerns, or if you have more questions?

---

### Official Review · Reviewer_p2T3 · 2023-07-24
**Potentially good and relevant dataset, but some important details missing and partially unclear validation**

**Rating:** 7
**Confidence:** 5

**Strengths:**

+ I agree with the authors that the absence of real-image datasets with ground truth poses is a limiting factor in the research of methods for tasks involving multi-view reconstructions. In this sense, the creation of this dataset is commendable and a step towards the right direction and a solid contribution towards having better data to evaluate novel methods.
+ Table 1 shows that the ground truth poses provided by the authors lead to better synthesis metrics than COLMAP poses, which is an indirect measure of the goodness of such ground truth poses.
+ The size of the dataset (36 objects and ~10K images) is reasonable for the proposed tasks. As the authors argue, their dataset can also be used to a wider variety of tasks than the ones validated in their paper.
+ The analyses in the paper (e.g., dependence of 3 NeRF methods with varying camera noise), in addition to being useful for validating the data, are also interesting and relevant.

**Additional Feedback:**

I would like to congratulate the authors on their work, and I would like to read their comments to my concerns.

**Clarity:**

The paper is mostly clearly written. In particular, the dataset creation steps and the validation tasks and results are very clear. As mentioned previously, my only concern is that the dataset specs should be introduced and compared with related datasets earlier in the paper and in a straightforward manner, ideally in the form of a table.

**Correctness:**

The dataset is created in a mostly sound manner, with the steps correctly detailed, and the experiments designed for validation are mostly illustrative of the use of the data. As reported in previous section, I have concerns on the manual alignment and its implications and the characterization of the error of the 3D shape ground truth.

**Documentation:**

As far as I checked, there is a project web with instructions to download the dataset, that is available. There is a project web and instructions to download and use the data in github. The data is in Google Cloud. The authors have a maintenaince plan in the supplementary material. There is a Jupyter notebook with a tutorial for an easy start. The data is released uncer CC-BY, and the code under Apache. In my opinion, the documentation and availability of the data is excellent in this case.

**Ethics:**

There is no ethical concerns for this paper. The data is recorded from the authors themselves and, as far as I checked, is object-centric with simple backgrounds without persons or human-identifiable features, hence without privacy- or consent-concerning content. I cannot see any legal concern either.

**Limitations:**

The authors include a discussion on limitations in Section 6, which I find reasonable. Negative societal impact is not discussed, however, I cannot see any potential negative impact to discuss here.

**Opportunities For Improvement:**

- While the goodness of the ground truth camera poses comes accross very clearly in Table 1, the quality of the 3D shape ground truth is unclear from the paper. The experiment on correspondence estimation, although it involves the 3D ground truth, does not inform about its quality. The authors should try to inform about the quality of the 3D scans more straightforwardly, e.g., adding the sensor specs instead of referencing the manufacturer web, or adding experiments in which its quality can be measured.
- The dataset construction steps are well described in the paper. However, the dataset characteristics in comparison with similar ones from the literature is unclear. Numbers for NAVI are spread over the paper, many times described just qualitatively, and for related datasets are mostly missing. A table compiling quantitative characteristics of NAVI and related datasets would be useful for assessing more easily the author's contribution.
- Manual 2D-3D alignment is questionable. Using rendering quality as a proxy in Table 1 (and qualitative results in the supplementary) suggests lower errors than COLMAP. However, a direct measure of the pose error is missing. Pose errors will be presumably variable, depending on the alignment quality and the geometry of the object and its relative position with the camera. Aggregating results might hide that some poses might be noisier than others. I do not have a clear suggestion for improvement for this, but it is a clear limitation of the dataset in my opinion.
- In Table 3, the caption explains 0.2 as "confidence". What is exactly confidence?
- Also in Table 3, why only rotation errors are reported, and not translation ones? What is the specific method used to estimate motion from correspondences? Details would be important here, for example, is RANSAC used?

**Relation To Prior Work:**

Up to my knowledge, the most relevant previous work is correctly reference and discussed, with the only caveat of being the discussions qualitative at some points. Quantifying the data specs with respect to the literature in a clear manner is adviced.

**Summary And Contributions:**

The submission presents a dataset composed of multiple views of objects in two setups (1 - controlled scene with no background/lighting/intrinsics variation and 2 - in-the-wild setup with varying background/lighting/intrinsics). The authors provide as (pseuo-) ground truth 1 - accurate 3D shape measurements (by professional 3D scanners) and 2 - accurate camera poses by manual 2D-3D alignment between the 3D scans and RGB images. The data is extensively validated in image synthesis and feature correspondence tasks.

---

> ### Author Response · Authors · 2023-08-22
> **Author response**
>
> ---
> ### **”Quality of 3D shape ground-truth is unclear”**
>
> The 3D scans were obtained by using [EinScan-SP](https://www.einscan.com/einscan-sp/einscan-sp-specs) and [EinScan Pro HD](https://www.einscan.com/handheld-3d-scanner/einscan-pro-hd/einscan-pro-hd-specs/). We used their fixed scan mode (accuracy of 0.05mm and 0.04mm, respectively). Additionally, we manually curated the mesh, removed noisy vertices, and posed them in a consistent pose (Y+ axis is up, Z- axis is front, like in ShapeNet. [[Figure A2](https://drive.google.com/file/d/1yqG242PU4r0KBak2j4p5EH4JKnsh9NcG/view?usp=drive_link) in the supplementary] illustrates the quality of our scans. We provide accurate shapes with very fine-grained details.
>
> The reviewer can additionally check the quality of all scans, and alignments in the following [slides](https://docs.google.com/presentation/d/1LCWUBQHs3oGi1bwCQjgLm8-etV9Y8ldPSpFN0BLoi6o/edit#slide=id.p)
>
> ---
> ### **More dataset Statistics**
>
> Please refer to the above General Response paragraph: **Overview of dataset statistics**.
>
>
> ---
> ### **”Manual 2D-3D alignment is questionable”**
>
> We now quantitatively evaluate the poses using inter-annotator agreement as the metric.
> Please refer to the above General Response paragraph: **2D-3D alignment quality and inter-annotator consistency** and also the above common response on **Annotation rationale**.
>
>
> ---
> ### **What is confidence in the correspondences table-2?**
>
> Several methods (eg. learned matchers) produce a matching confidence in the form of soft assignment matrix for their predicted correspondences, with values in the range of [0.0, 1.0].
> We threshold the matching confidence, and ignore all estimated correspondences with confidence values below 0.2. This is standard practice in the learnable matching literature. For some methods (e.g. mutual nearest neighbor), there is no confidence parameter and we simply use all proposed correspondences. We added a short explanation on Table 5.
>
>
> ---
> ### **”Why only rotation errors are reported and not the translation ones?”**
>
> We follow the standard established by the learnable matching literature, e.g. SuperGlue and LoFTR, which both report relative pose estimation results on rotation only, as pose predictions are up-to-scale in other datasets.
>
> ---
> ### **What is the specific method used to estimate motion from correspondences?**
>
> To estimate motion from correspondences, we use RANSAC for essential matrix estimation. Given the predicted correspondences and known camera intrinsics, we sample a minimum set of correspondences to calculate the essential matrix and calculate the number of inlier correspondences to this predicted E-matrix. This process is repeated many times, and we keep the E-matrix with the most inliers as our final estimation. From the E-matrix, we decompose this into rotation and translation matrices.

---

> > ### Comment · Reviewer_p2T3 · 2023-08-25
> > **Raised the rating to 7, good paper, accept, as the authors' replies solved my main concerns**
> >
> > Thank you for your careful and well motivated answers to my concerns. With the added discussion and material, I am more convinced of the contribution of the dataset, and hence raised my rating to 7, good paper.
> >
> > Specifically, I know see more clearly the relevance of the dataset compared to existing ones, thanks to the new Table 3. The quality of the ground truth shape is now clear with your reply, I suggest to highlight this more in the paper. On the quality of the 2D-3D alignment, the analysis made by the authors is sufficient to me, as it shows that the variability between annotators is much smaller than the errors of the methods, and then it can serve as ground truth. My suggestion here is to explain this in the paper making this comparison between annotator variability and typical errors. 1.7 degrees on itself does not qualify your ground truth as good or bad. It is the relation of this 1.7 degrees with the magnitude of the typical errors of your methods, which are of several tens of degrees in Table 4, that motivates its use as ground truth.
> >
> > I think details on the confidence and motion estimation are important. I would even add more details in the motion estimation. I suppose you use Nister's 5 points, write it explicitly. BTW, a note here, I think keeping the Essential matrix with the highest consensus does not give you the best estimate you coud have. My suggestion: re-estimate the Essential matrix with **all** the correspondences in consensus with the most voted one, instead of taking the most voted one that has been estimated only from 5 correspondences. It should give you more accurate results.

---

> > > ### Author Response · Authors · 2023-08-26
> > > **Thanks for the feedback**
> > >
> > > Thank you again for your constructive feedback. We are glad that our rebuttal addressed the raised issues. We will include the proposed suggestions in the revised version of the paper. Specifically:
> > > - We will put the rater (dis-)agreement of our dataset into perspective, by comparing it to automatic methods. This is an excellent idea, which will highlight how much room for improvement exists while this dataset serves as ground truth.
> > > - We will include an explicit statement that we used Nister’s 5-point algorithm. We actually already re-estimate the essential matrix exactly as the reviewer suggests: by including all inliers to the essential matrix with the highest consensus (from standard RANSAC).
> > > - If space permits, we will move Fig. A2 that highlights the quality of the scans to the main paper, after incorporating the comments from all reviewers.

---

### Author Response · Authors · 2023-08-22
**List of Changes to the paper and the supplementary**

Following reviewers suggestions, the new additions mainly comprise of dataset and annotation analysis; and tables comparing NAVI with existing datasets. Specifically, we make the following changes:

#### **Changes in the main paper**:
- New subsection (2.2) in the main paper (lines-130-169) for ‘Dataset analysis’ with the following paragraphs:
	- General statistics, including Table 1.
	- Annotation quality analysis.
	- Manual vs. Semi-automatic alignment process.
- On using self-captured vs. internet images.
- Moved the table of comparisons to the existing in-the-wild datasets from the supplementary to the main paper (Table-3 in page-6).
---
#### **Changes in the supplementary**:
New visualizations to demonstrate the high quality of the NAVI shapes and annotations.
Figure A2 (Subsection 1.2)  shows the high quality of scans.
Figure A3 (Subsection 1.3) presents the issues with semi-automatic annotation.
Figure A4 (Subsection 1.4) iIllustrates why we captured images ourselves instead of using internet images.
- New table listing all camera devices used in NAVI (Table A1).
- New subsection 2.1, and Table A2 in page 5, comparing NAVI to multiview datasets.

---

### Author Response · Authors · 2023-08-22
**Annotation Rationale (all reviewers)**

**Why did we scan our own 3D models instead of using existing models, say ShapeNet?**:

See bullet point 1, 2, and 3 in the comment below.

---
**Why did we capture images on our own instead of using readily-available internet images?**:

See point 2.d, 2.e, and 3 below.

---
**Why did we use manual 2D-3D alignment instead of semi-automatic approaches such as 2D-3D point correspondences?**:

See point 2.a, 2.b, 2.c, and 3 below.

---
**How we ended up with the current annotation pipeline?**

Our journey started several years ago with an ambitious goal of creating a large-scale dataset by aligning 3D models and corresponding images from the internet, but as we discovered many technical challenges along the way, we adapted as follows:

1. We started with **ShapeNet** (~50k) models and mined **Internet images**. We selected images based on their similarity to ShapeNet model renderings and performed automatic alignment based on silhouette matching, but we realized that without exact object matches, good alignment is impossible to achieve.
2. We switched to [**Google Scanned Objects**](https://ai.googleblog.com/2022/06/scanned-objects-by-google-research.html) (~1k) because each object is associated with a product name, which we used for mining **Internet images**.
- a. Even with matching objects, we found **automatic silhouette matching**, which was highly sensitive to its initial conditions, still fails to produce high quality alignment in nearly all cases.
- b. We also tried utilizing **2D-3D point correspondences** as a fast annotation process, but the resulting alignment is not precise enough for our intended downstream tasks ([Figure A3](https://drive.google.com/file/d/1Bc7VYW0dmKmDwZA-1hxjHda3k_avYlMR/view?usp=drive_link) left result).
- c. We ended up focusing on developing an independent **manual 2D-3D alignment**  pipeline, drawing inspiration from [Pascal3D+](https://cvgl.stanford.edu/projects/pascal3d.html) and hoping to achieve superior alignment quality ([Figure A3](https://drive.google.com/file/d/1Bc7VYW0dmKmDwZA-1hxjHda3k_avYlMR/view?usp=drive_link) right result).
- d. However, precise alignment was still extremely difficult to achieve due to two compounding factors: 1. Unknown camera intrinsics (e.g. focal length) for Internet photos and, 2. Uncertainty that the 3D model matches the actual object captured in the photos.
- e. To decouple them, we tried **capturing photos using our own cameras with known intrinsics**. This is when we discovered that most 3D models do not exactly match the objects captured in Internet photos due to deformation, moving parts, and slight differences in manual facturing processes (See [Figure A4](https://drive.google.com/file/d/1PbrNRVYa-e9RzGWwe0acKZiQVu3aOXC-/view?usp=drive_link)). We observed a similar phenomenon in [Pix3D](http://pix3d.csail.mit.edu/) and [ABO](https://amazon-berkeley-objects.s3.amazonaws.com/index.html).

3. Therefore, we decided to **scan our own 3D models** and align them with the photos we capture ourselves. Finally, we were able to train our human annotators to produce near-perfect 2D-3D alignment ([Figure A3](https://drive.google.com/file/d/1Bc7VYW0dmKmDwZA-1hxjHda3k_avYlMR/view?usp=drive_link) right result). For a quantitative comparison, we annotated the same 30 randomly sampled images with point correspondences and measured inter-annotator agreement between two rounds of annotation. The average 3D rotation distance was 6.5 degrees, and the average 3D translation distance was 7.02mm, significantly higher than our alignments (1.7 degrees, and 0.97mm, respectively). We added this rationale in Section-2.2 of the revised paper (L136-161).

In summary, our final NAVI annotation pipeline is a rather *deliberate* process that took us over 2 years to nail down, not the result of random design choices. As a result, NAVI provides high-quality 2D-3D aligned image collections that enables evaluations that are not possible with existing datasets.

---
**On expanding the dataset**

Equipped with a firm understanding of this process, we are making progress on relaxing certain constraints. For example, we are currently training our annotators to perform alignment tasks with unknown camera focal length. We found that when a human annotator is certain about exact-matching-3d-model-and-depicted-objects, they could be trained to excel at estimating focal length with the help of our manual 2D-3D alignment tool.

Other efforts we are working on include 1) scanning more objects and 2) annotating entire video sequences with frame sampling and tracking. We are confident to make the commitment to expanding the dataset over time, as suggested by both reviewer vLyG and eykt.

---

### Author Response · Authors · 2023-08-22
**General Response**

---
### **Comparison to other related datasets** (rev. p2T3, eykt, Ymgi)

We provide both multi-view image collections, and in-the-wild image collections (with varying illuminations and cameras). Table 3 in the revised paper compares NAVI with other in-the-wild image collections. NAVI provides a significantly larger number of image collections and it is the only dataset with GT shapes and poses. We show the table here for convenience:

*Table 3: Comparing NAVI with existing datasets with in-the-wild image collections, where the task is 3D reconstruction of an object with images captured in different backgrounds and illuminations.*
| Dataset        | #Objects | #Scenes | #Images | Camera Poses | 3D Annotations | 2D-3D Alignment |
| :-----------   | :----------: | :-------: | :------------: | :---------: | :-----: | :------: |
| LASSIE         | 6           | 6          | 180           | -              | Keypoints           | No          |
| E-LASSIE       | 6           | 6          | 270           | -              | Keypoints           | No          |
| NeRD           | 8           | 8          | 396           | Synthetic GT   | -                   | No          |
| NeRF-W         | -           | 6          | 7658          | -              | -                   | No          |
| SAMURAI        | 8           | 8          | 560           | -              | -                   | No          |
| NeROIC         | 3           | 3          | 132           | COLMAP         | -                   | No          |
| NAVI (Ours)     | 36          | 36         | 2298          | Near GT        | Scanned mesh        | Yes         |


Table A2 in the supplementary compares NAVI with multiview datasets. Several multiview datasets are larger than NAVI. However, the key aspect of NAVI is that none of the existing real-world datasets (e.g. CO3D or MVImgNet) provide high-quality pose and shape annotations from different cameras and in different environments.


---
### **On the number of objects (size) in the dataset** (rev. eykt, vLyG, kSQG)

  One of NAVI’s primary use-cases is training and evaluating 3D reconstruction from multi-view and in-the-wild image collections (Sections 3 and 4 of main paper), where training and evaluations are done *within* an image collection. We believe 36 objects in NAVI provides a good diverse set of objects for robust evaluation of 3D reconstruction techniques (reviewer p2T3 acknowledges this). For instance, most existing works on neural radiance fields evaluate on <10 image collections (See Table 3 above).
  For correspondence estimation (Section 5), NAVI enables *evaluating* features densely in real images. Our dataset is not suitable for large-scale training of correspondence estimation networks, and this is not the intended usage. We provide a high-quality real-image benchmark which unlocks new ways of evaluating models, and reveals their limitations (Table 5).


---
### **Overview of dataset statistics** (rev. p2T3, eykt, vLyG, Ymgi)
Thanks for the suggestion.
We added a paragraph and a table on dataset statistics in the main paper (L131-135, Table 1).
We listed all devices used to capture the images (Table A1, supplementary).


---
### **2D-3D alignment quality and inter-annotator consistency** (rev. p2T3, kSQG, Ymgi)

We present a way to quantitatively evaluate our alignments in Subsection 2.2, paragraph “Annotation quality analysis” of the main paper. We copy the text here:

To analyze the quality of our 2D-3D alignments, we annotated 30 randomly selected images with two different annotators and measured the inter-annotator agreement [A] using two metrics: 3D translation distance (in milimeters), and 3D rotation distance (in degrees) between the two alignments. The average 3D rotation distance between two verified alignments is 1.7 degrees, and the 3D translation distance is 0.97 milimeteres. The very small differences in the obtained alignments from two independent raters highlight the high quality of the alignments in the NAVI dataset. Similarly, we measured the quality of the alignments that we reject as *wrong*, by comparing them to alignments of the same image that were verified as *correct*. In this case, average annotator disagreement is 2.3 degrees of 3D rotation, and 2.01 milimeters of 3D translation. Practically, this means that even slight mis-alignments did not pass our strict verification process.

[A] Computing Krippendorff's Alpha-Reliability


---
### **On releasing the annotation tool** (rev. vLyG,  kSQG)

We plan to release the annotation tool, after decoupling it from internal software.

---

### Decision · Program_Chairs · 2023-09-22

**Decision:**

Accept (Poster)

**Comment:**

All reviewers except one agree that the paper should be accepted. The reviewer opposing acceptance does not have major concerns, they are related with limited size and limited applications, which are concerns I recognize. However, after reading the rebuttal and discussions, I do believe that the paper and dataset is valuable for community, and authors promise to expand its size in the future. Hence, I propose to accept the paper.